# A ribosome-associated chaperone enables substrate triage in a cotranslational protein targeting complex

Hao-Hsuan Hsieh [1], Jae Ho Lee[1,2], Sowmya Chandrasekar[1] & Shu-ou Shan [1]✉

Protein biogenesis is essential in all cells and initiates when a nascent polypeptide emerges from the ribosome exit tunnel, where multiple ribosome-associated protein biogenesis factors (RPBs) direct nascent proteins to distinct fates. How distinct RPBs spatiotemporally coordinate with one another to affect accurate protein biogenesis is an emerging question. Here, we address this question by studying the role of a cotranslational chaperone, nascent polypeptide-associated complex (NAC), in regulating substrate selection by signal recognition particle (SRP), a universally conserved protein targeting machine. We show that mammalian SRP and SRP receptors (SR) are insufficient to generate the biologically required specificity for protein targeting to the endoplasmic reticulum. NAC co-binds with and remodels the conformational landscape of SRP on the ribosome to regulate its interaction kinetics with SR, thereby reducing the nonspecific targeting of signalless ribosomes and preemptive targeting of ribosomes with short nascent chains. Mathematical modeling demonstrates that the NAC-induced regulations of SRP activity are essential for the fidelity of cotranslational protein targeting. Our work establishes a molecular model for how NAC acts as a triage factor to prevent protein mislocalization, and demonstrates how the macromolecular crowding of RPBs at the ribosome exit site enhances the fidelity of substrate selection into individual protein biogenesis pathways.

[1] Division of Chemistry and Chemical Engineering, California Institute of Technology, Pasadena, CA 91125, USA. [2]Present address: Department of Biology, Stanford University, Stanford, CA 94305, USA. ✉email: sshan@caltech.edu

Generation and maintenance of a functional proteome requires the proper biogenesis of all the newly synthesized proteins, a process that often begins before nascent proteins finish their synthesis. Upon emergence from the ribosome tunnel exit, a nascent chain (NC) becomes accessible to a variety of ribosome-associated protein biogenesis factors (RPBs) that share overlapping docking sites near the tunnel exit (Fig. 1a).

These RPBs direct the nascent polypeptide to distinct biogenesis pathways including localization to cellular membranes[1], folding[2], maturation[3], and quality control[4]. How a NC recruits the correct set of RPBs and thus commits to the proper biogenesis pathway in a timely manner is unclear. How multiple RPBs coordinate with one another in the crowded space near the ribosome tunnel exit is also an unanswered question, especially on eukaryotic

**Fig. 1 Selective protein targeting by SRP in translation lysate. a** Projections of the EM densities of structurally resolved RPBs onto the surface of the 80S ribosome (PDB: 4UG0; light gray)[78] viewed from the tunnel exit (marked by an asterisk). The electron density map of the RPBs (SRP: EMD-3037[6], green; NAC: EMD-4938[23], magenta; RAC: EMD-6105[79], red; and NatA/E: EMD-0202[80], blue) were overlayed using the structure of 80S in Chimera (UCSF). The densities of ribosomal proteins surrounding the tunnel exit are shown in different shades of gray, with dotted areas indicating the parts buried underneath the ribosomal surface. Due to the limited resolutions of the original EM maps, the projections of NAC and RAC are likely incomplete. **b** Composition of the signal sequence variants tested in this work. **c** Translocation assay showing the specific targeting of pPL(wt) and pPL(ss). SRP-dependent cotranslational protein translocation were measured for the indicated nascent proteins as described in the Methods section and analyzed by SDS-PAGE and autoradiography. The bands for preprolactin (pPL) and signal sequence-cleaved prolactin (PL) are indicated. **d** Quantification of the data in **c**. The values for pPL(ssmt) and pPL(ssmt2) are very close and overlap with each other in the figure. Translocation efficiency was calculated using Eq. (3) in the Methods section. Source data for **c** and **d** are provided in the Source Data file.

ribosomes where mechanistic information is limited for many RPBs[5]. Here, we provide insight into these questions by investigating how a cotranslational chaperone, the nascent polypeptide-associated complex (NAC), regulates the conformation and activity of signal recognition particle (SRP) to enable substrate triage during cotranslational protein targeting.

The universally conserved SRP and its receptor (SR) couple the synthesis of ~30% of the newly synthesized proteome to their localization at the eukaryotic endoplasmic reticulum (ER), or the prokaryotic plasma membrane[1]. SRP recognizes ribosome-nascent chain complexes (RNCs) with a transmembrane domain (TMD) or signal sequence on the NC and, via interaction with SR, delivers the RNCs to the Sec61p translocase at the ER (or SecYEG at the prokaryotic plasma membrane). Eukaryotic SRP consists of a 7SL SRP RNA tightly bound to six proteins (SRP9, SRP14, SRP19, SRP54, SRP68, and SRP72). The universally conserved SRP54 interacts with the 5.8S rRNA, uL29, and uL23 near the ribosome exit site via a helical N-domain and recognizes the hydrophobic TMD or signal sequence on the NC via a methionine-rich M-domain[6]. The eukaryotic SR is an SRα/β heterodimer anchored at the ER membrane. Both SRP54 and SRα contain homologous GTPase, NG-domains that stably dimerize with one another upon GTP binding, thus delivering SRP-bound RNCs to the ER surface[7–9]. SRP and SR undergo significant conformational changes upon their assembly, which culminates in reciprocal GTPase activation that drives their disassembly and recycling[10–12].

Eukaryotic SRP must specifically recognize and target the ribosomes translating proteins destined to the endomembrane system, while rejecting those destined to the cytosol, nucleus, and other cellular membranes such as mitochondria[13]. The molecular mechanism underlying this selectivity is poorly understood. While earlier models suggested that SRP binds weakly to RNCs without a strong signal sequence, mammalian SRP was found to bind tightly to RNCs with or without a signal sequence, with equilibrium dissociation constants ($K_d$) of <10 nM[14,15]. These observations suggested that even signalless ribosomes could be bound by SRP at its in vivo concentration (~500 nM)[16]. Recent ribosome profiling and cryo-electron microscopy (cryo-EM) structures also suggested that eukaryotic SRP can bind to RNCs before the signal sequence emerges from the ribosome exit tunnel[6,17]. The discrepancy between the biological selectivity required for targeting and the low specificity observed in SRP-RNC binding could be explained by molecular events after RNC-SRP binding, such as selective activation of SRP-SR binding by RNCs with a signal sequence and/or proofreading through GTP hydrolysis[18]. While this was the case in the bacterial SRP pathway[18], recent studies showed that RNC with a signal sequence accelerates complex formation between human SRP and SR only 10–20 fold faster than the empty ribosome or RNC without a functional signal sequence for ER targeting (termed signalless ribosomes), whereas the corresponding difference is 200–3000 fold with bacterial SRP and SR[10]. These results suggest that mammalian SRP and SR need regulation by additional 'triage' factors.

A strong candidate for such a factor is NAC, an abundant chaperone conserved throughout eukaryotic organisms. NAC is a heterodimer of NACα and NACβ subunits, which dimerize through their NAC domains to form a β-barrel-like structure[19,20]. NAC is present at similar abundance as ribosomes in the cytosol[16,21], associates with a variety of translating ribosomes[22], and can crosslink to the NC when the latter is still inside the ribosome exit tunnel[23]. Deletion of NAC causes synthetic protein aggregation phenotype with the deletion of another cotranslational chaperone, Ssb, in yeast[24] and is lethal in higher eukaryotes[25,26], implicating it in cotranslational nascent protein

folding. While the precise cellular functions and biochemical activities of NAC remain to be determined, multiple evidence suggest that NAC serves as a negative regulator of protein targeting to the ER. Early works in rabbit reticulocyte lysate (RRL) showed that NAC reduces non-specific, salt-sensitive crosslinks of SRP to signalless nascent chains on the ribosome[27–29]. In addition, microarray analysis in yeast showed that the deletion of NAC alters the spectrum of RNCs associated with SRP[22]. Finally, NAC depletion led to the mistargeting of signalless RNCs to the ER in both in vitro targeting assays and in vivo[21,25,27,28,30]. However, the mechanism by which NAC enhances targeting selectivity remains unknown. Crosslinking[31,32] and cryo-EM analyses[23] showed that NAC docks at the ribosomal protein uL23 near the exit site, which overlaps with the ribosome binding sites of SRP54-NG and the Sec61p translocon. It was therefore suggested that NAC blocks the nonspecific binding of SRP or Sec61p to the ribosome[25]. Neither model had definitive evidence due to the lack of assays that accurately resolve and measure molecular events in the targeting pathway, nor was it clear whether these mechanisms contribute significantly to the enhancement of specificity during protein targeting.

In this work, we used quantitative biochemical and biophysical measurements to dissect when and how NAC regulates the mammalian SRP pathway. We developed quantitative binding assays to measure the interaction of RNC with individual RPBs, which showed that SRP and NAC can co-bind on RNCs with low nanomolar affinity and modest anti-cooperativity. Both total RPBs reconstituted from cell lysates, as well as purified NAC selectively reduce the rates of SRP-SR association on signalless RNCs and RNCs with a buried signal sequence. Single-molecule measurements show that NAC regulates SRP-SR association by selectively biasing the conformational landscape of ribosome-bound SRP. These data allowed us to construct an analytical mathematical model for cotranslational protein targeting, which demonstrates that the NAC-induced regulation of SRP-SR assembly kinetics is essential for the specificity of SRP-dependent targeting under in vivo-like conditions. Our work establishes a model for how NAC acts as a triage factor for the SRP pathway and provides valuable concepts and tools to understand nascent protein selection and triage on the ribosome.

## Results

**Ribosome-associated factors enhance the specificity of RNC-activated SRP-SR association.** To evaluate the selectivity of SRP-dependent protein targeting, we used an established cotranslational targeting assay[33] that measures the ability of purified human SRP and SR to mediate the translocation of preproteins translated in the wheat germ lysate (which lacks endogenous SRP) into ER microsomes stripped of endogenous SRP and SR. As a model SRP substrate, we used the preprolactin (pPL) nascent chain (Fig. 1b). Both wildtype (wt) pPL and a variant in which the pPL signal sequence was replaced by a synthetic signal sequence (ss; Fig. 1b) were efficiently translocated by SRP and SR into ER microsomes, as evidenced by signal cleavage of pPL to prolactin (PL; Fig. 1c). Introduction of two charged residues into the pPL signal sequence (ssmt) or replacement of the signal sequence with part of the cytosolic domain of Sec61β (ssmt2) abolished translocation (Fig. 1b–d). The absence of signal sequence cleavage product indicates that cotranslational translocation mediated by SRP effectively rejects nascent proteins with a weak or no signal sequence in a complete cell lysate.

To decipher the molecular basis of this targeting specificity, we generated RNCs bearing the first ~90 amino acids of pPL(ss), pPL(ssmt), and pPL(ssmt2) (Supplementary Fig. 1a). To obtain monosomes stripped of peripherally bound RPBs, the RNCs were

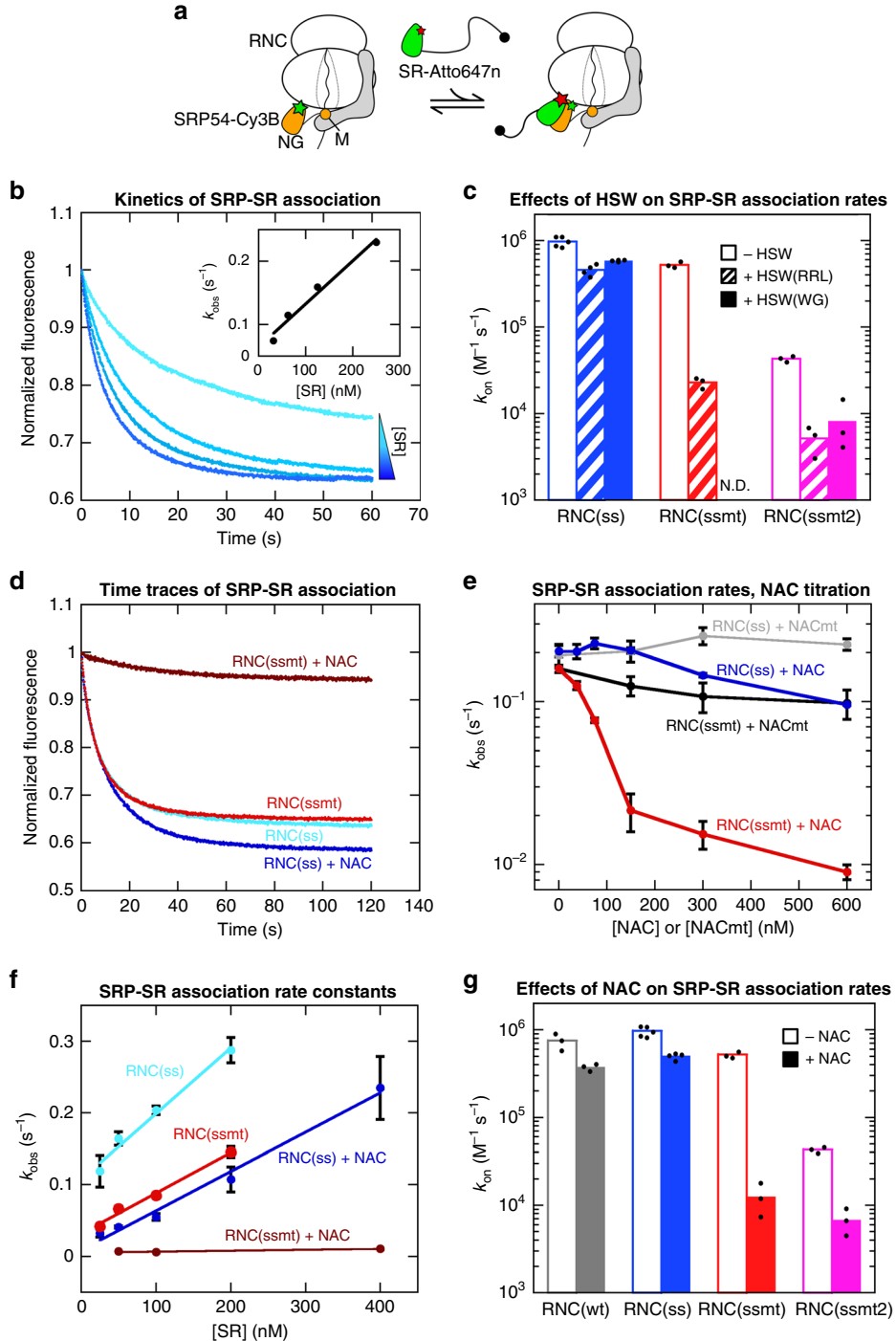

purified through a high salt (1 M KOAc) sucrose cushion, affinity purification via an N-terminal 3× FLAG tag on the nascent chain (Supplementary Fig. 1a), and sucrose gradient centrifugation[34]. As SRP-SR binding is activated $10^2–10^3$ fold by RNCs[10], we first measured this binding using an assay based on Förster resonance energy transfer (FRET) between donor (Cy3B) and acceptor (Atto647n) dyes labeled on the NG-domains of SRP54 and SRα, respectively (Fig. 2a)[10]. These measurements used a soluble SRαβΔTM lacking the nonessential N-terminal TMD of SRβ (abbreviated as SR). The soluble SR can support yeast growth[35], is fully functional in supporting protein translocation (Fig. 1)[36], and has the same GTPase activity as wildtype SR[12]. We followed the kinetics of SRP-SR assembly as a function of SR concentration to determine their association rate constant ($k_{on}$) (Fig. 2b). SRP-

SR association on RNC(ss) was rapid, with a $k_{on}$ of ~$10^6 M^{-1}s^{-1}$ (Fig. 2c, first column and ref. [10]). However, SRP-SR association on RNC(ssmt) was only two-fold slower than on RNC(ss), with a $k_{on}$ of $5 \times 10^5 M^{-1}s^{-1}$ (Fig. 2c), in strong contrast to the effective rejection of this substrate in the protein translocation reaction (Fig. 1).

To test if additional components in the cell lysate were responsible for this discrepancy, we extracted ribosome-associated proteins using a high salt wash (HSW) of the RRL ribosome. After adjusting the ionic strength in HSW to physiological conditions (150 mM KOAc), the effect of HSW on RNC-activated SRP-SR assembly was determined. The presence of HSW(RRL) reduced the SRP-SR association rate constant on RNC(ssmt) 20-fold without substantially affecting

**Fig. 2 NAC is sufficient to increase the specificity of SRP-SR association. a** Scheme of the FRET assay to measure the interaction between SRP (gray with SRP54 shown in orange, NG-domains and M-domains of SRP54 are indicated) and SR (green). Green and red stars denote the donor and acceptor dyes, respectively. **b** Representative fluorescence time traces of SRP-SR association, measured as described in the Methods section. The fluorescence signal was normalized to the intensity at time 0 in each trace. The time traces were fit to an exponential decay function to obtain the observed rate constant, $k_{obs}$, at each SR concentration. The inset shows the plot of $k_{obs}$ against SR concentration, which was fit to Eq. (1) to obtain $k_{on}$, the association rate constant for SRP-SR binding. The range of [SR] is from 31.3 to 250 nM. **c** Summary of the effects of HSW on the $k_{on}$ values for SRP-SR binding on RNC(ss), RNC(ssmt), and RNC(ssmt2). The +HSW(RRL) and +HSW(WG) reactions contained RPBs obtained from 200 nM RRL or wheat germ ribosome, respectively. N.D., not determined. **d** Representative time traces of SRP-SR association on RNC(ss) and RNC(ssmt) in the absence and presence of NAC. All measurements contained 200 nM RNC, 10 nM SRP, 300 nM SR, and 300 nM NAC where indicated. Fluorescence signal was normalized to the intensity at time 0 in each measurement. **e** Dose-dependent effects of NAC on the apparent rate constants of SRP-SR association on RNC(ss) and RNC(ssmt). The effects of the ribosome binding mutant of NAC (NACmt) are shown in gray and black. The reactions contained 200 nM RNC, 10 nM SRP, 300 nM SR, and indicated concentrations of NAC or NACmt. **f** Observed association rate constants for SRP-SR binding ($k_{obs}$) were plotted against SR concentration and fit to Eq. (1) to obtain values of $k_{on}$. The reactions contained 200 nM RNC, 10 nM SRP, indicated concentrations of SR, and 300 nM NAC where indicated. **g** Summary of the effects of NAC on SRP-SR association rate constants on RNC(wt), RNC(ss), RNC(ssmt), and RNC(ssmt2). All values are shown as mean ± SD or mean with individual data points, with $n = 3$–5 independent measurements on the same biological sample. Source data for **b**–**g** are provided in the Source Data file.

the reaction on RNC(ss), increasing the difference in $k_{on}$ between RNC(ss) and RNC(ssmt) by over 10-fold (Fig. 2c, open vs. striped columns). HSW(RRL) also significantly reduced the rate of SRP-SR association on RNC(ssmt2), increasing the difference in $k_{on}$ between RNC(ss) and RNC(ssmt2) from 20-fold to 100-fold (Fig. 2c). HSW prepared from the wheat germ lysate (WG), which was used in assays of SRP-dependent cotranslational protein targeting (Fig. 1), showed similar effects to HSW(RRL) (Fig. 2c, solid columns). These results suggest that the specificity of cotranslational protein targeting is due, at least in part, to the presence of ribosome-associated factors, which selectively reduce SRP-SR association on RNCs without a functional signal sequence.

**NAC is sufficient for the specificity enhancement.** The abundant cotranslational chaperone NAC is a negative regulator of protein targeting to the ER[21,25,27,28,30]. To test whether NAC is responsible for the specificity enhancement observed with the HSW, we recombinantly purified the human NAC complex (Supplementary Fig. 1b). Titration of NAC in the SRP-SR association assay showed that low doses of NAC reduced the kinetics of SRP-SR binding on RNC(ssmt) by over an order of magnitude, whereas the RNC(ss)-activated SRP-SR association was largely unaffected by NAC across the titration range (Fig. 2d, e). The rate reduction with RNC(ssmt) was largely complete at NAC concentrations above 150 nM, close to the RNC concentration (200 nM) in this experiment, suggesting that NAC binds tightly and with equal stoichiometry to RNC(ssmt)[16,21]. A mutant NAC, in which the ribosome binding motif in NACβ was disrupted (NACmt: [27]RRK[29] to [27]AAA[29])[23,25,31], had negligible effects on SRP-SR association with both RNCs (Fig. 2e, black and gray), indicating that the interaction of NAC with the ribosome is important for its regulatory effect on SRP and SR.

To compare the effect of NAC to that of HSW, we measured the SRP-SR association rate constants at a saturating concentration of NAC (300 nM) (Fig. 2f, g). For comparison, quantitative Western blot analyses showed that HSW(RRL) contributed 150–250 nM NAC in the SRP-SR association measurements in Fig. 2c (Supplementary Fig. 1c, sum of the two splice variants[28]). Interestingly, although NAC is only one of the many ribosome-bound factors, the effects of NAC are the same, within error, as those observed with HSW(RRL), increasing the specificity of SRP-SR association on RNC(ss) relative to RNC(ssmt) from 2-fold to 40-fold (cf. Fig. 2g vs. 2c). NAC also increased the specificity of SRP-SR association on RNC(ss) relative to RNC(ssmt2) from 20-fold to 100-fold, similar to the effects of HSW(RRL) (Fig. 2g vs. 2c). Further, RNC(wt) has roughly the same $k_{on}$ as RNC(ss) with or without NAC (Fig. 2g), indicating that the effect of NAC

can be generalized to different signal sequences. Thus, NAC selectively slows down the recruitment of SR to SRPs on signalless RNCs and is sufficient to account for most of the specificity enhancement observed with HSW(RRL). These observations further indicate that NAC can regulate cotranslational protein targeting at a much earlier stage than the docking of ribosomes on the Sec61p translocase, as proposed previously[25,27,30,37].

**NAC and SRP co-bind tightly on RNCs.** The effects of NAC described above can be explained by two mutually exclusive models: (i) NAC and SRP compete with one another for binding the RNC. By excluding SRP from RNCs with no or weak signal sequences, NAC could inhibit the RNC-induced activation of SRP-SR assembly on SRP-independent substrates; (ii) NAC and SRP co-bind on the same RNC to form a ternary RNC-NAC-SRP complex, in which NAC induces conformational changes in SRP to regulate its interaction with SR.

To distinguish between these models, we developed FRET-based assays to quantitatively measure the binding affinity of RPBs for RNCs. To label the RNC, we used amber suppression based on an engineered pyrrolysine-tRNA/tRNA synthetase (PyltRNA/RS) pair from *Methanosarcina mazei* (*Mm*), which incorporates a clickable non-natural amino acid, axial-trans-cyclooct-2-en-L-Lysine (TCOK), into the nascent polypeptide at an amber codon during in vitro translation in RRL. TCOK undergoes Diels-Alder reactions with tetrazine-conjugated fluorophores[38], allowing site-specific incorporation of a fluorescent probe into the nascent protein. We chose this system because of the well-established bio-orthogonality of *Mm*PyltRNA/RS in mammalian cells and the rapid, specific reaction of a strained alkene with tetrazine[39]. The efficiency of amber suppression using this system is ~80% under optimized conditions (Supplementary Fig. 2). We also observed efficient and specific labeling of TCOK-containing RNCs with tetrazine-conjugated BODIPY-FL (BDP), with negligible off-target labeling of nascent chains that do not contain TCOK (Supplementary Fig. 3).

Using this method, we incorporated BDP as the donor dye one residue upstream of the signal sequence in RNC(ss) and RNC(ssmt). As the FRET acceptor dye, we labeled the SRP54 NG-domain at residue 12 with tetramethylrhodamine (TMR) using thiol-maleimide chemistry[10]. The estimated distance between the dye pair is <35 Å based on available structures[6]. Incubation of RNC[BDP] with SRP[TMR] resulted in a reduction in donor fluorescence intensity and a corresponding increase in the fluorescence intensity of the acceptor dye (example for RNC(ssmt) in Fig. 3b). Both changes were reversed by the addition of excess unlabeled SRP, indicating that the observed fluorescence

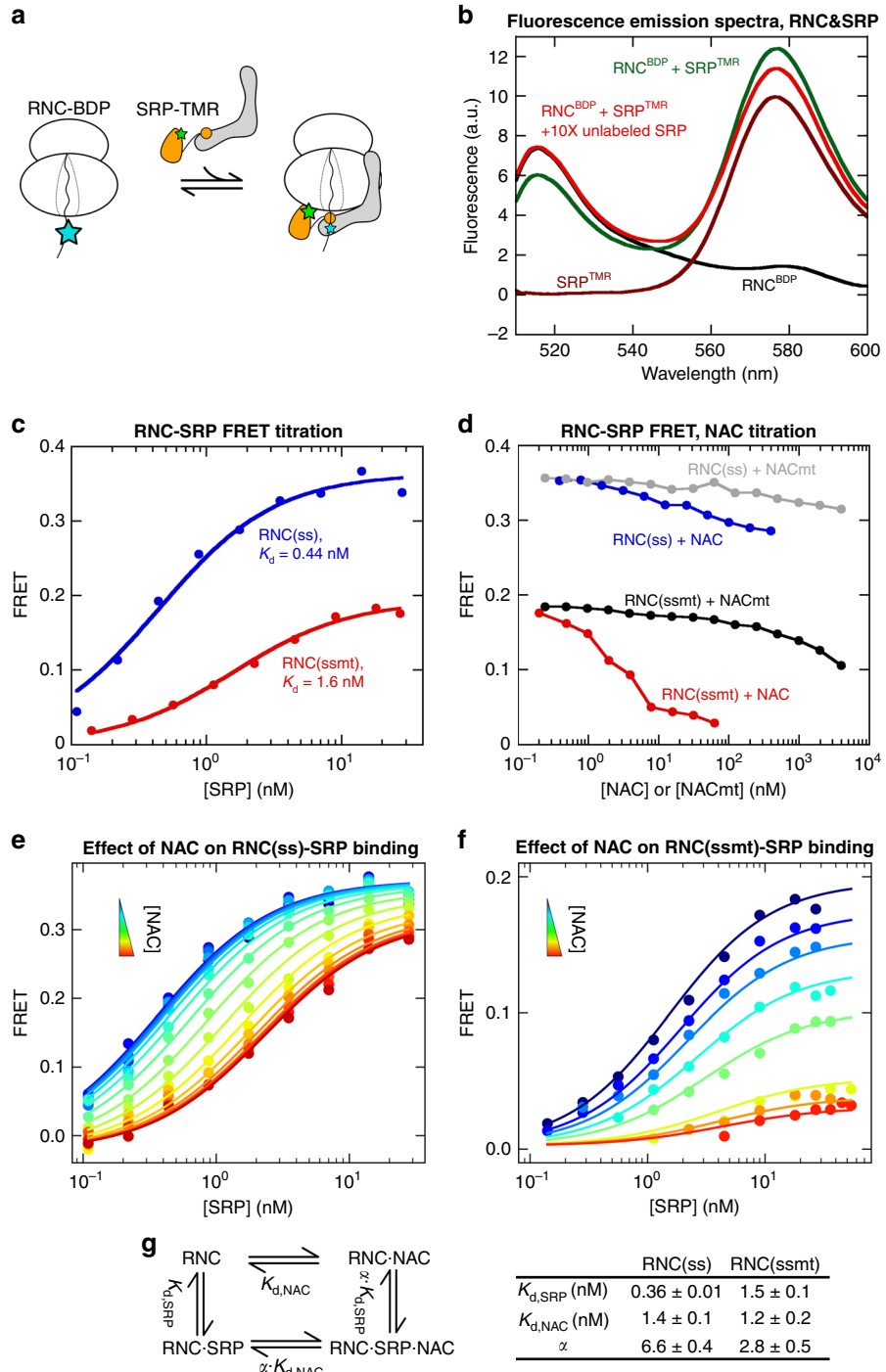

**Fig. 3 An RNC-SRP binding assay shows that NAC co-binds with SRP on the ribosome. a** Scheme of the FRET assay to measure RNC-SRP binding. RNC was labeled with BDP (cyan star) at the N-terminus of signal sequence, and SRP was labeled with TMR (green star) at residue 12 of SRP54. **b** Fluorescence emission spectra showing FRET between RNC(ssmt)[BDP] and SRP[TMR], using an excitation wavelength of 485 nm. Where indicated, the reactions contained 1 nM RNC(ssmt)[BDP], 5 nM SRP[TMR], and 50 nM unlabeled SRP. **c** Equilibrium titrations to measure the $K_d$ of the RNC-SRP complex. Titrations contained 1 nM BDP-labeled RNC(ss) or RNC(ssmt) and indicated concentrations of SRP[TMR]. FRET efficiency was calculated using the fluorescence emission intensity at 517 nm according to Eq. (2). The lines are fits of the data to Eq. (4), which gave the indicated $K_d$ values. **d** Effects of NAC and NACmt on the FRET efficiency between RNC[BDP] and SRP[TMR]. The reactions contained 1 nM BDP-labeled RNC(ss) or RNC(ssmt), 20 nM SRP[TMR], and indicated concentrations of NAC or NACmt. **e, f** RNC-SRP FRET titrations in the presence of increasing NAC concentrations for RNC(ss) (**e**) and RNC(ssmt) (**f**). The titrations contained 1 nM RNC[BDP], indicated concentrations of SRP[TMR], 0–400 nM NAC (**e**) or 0–62.5 nM NAC (**f**). The lines are global fits of the data to the model in **g** using Eq. (8) in the Methods section. **g** Left panel: model for the coupled binding of SRP and NAC to the RNC. The same coupling factor α describes the degree to which SRP affects the RNC binding affinity of NAC and vice versa, as dictated by the principle of thermodynamic coupling. Right panel: summary of the parameters obtained from global fits of the data in **e** and **f** to Eq. (8). All values are reported as optimized value ± square root of covariance (equivalent to fitting error). Source data for **b**–**f** are provided in the Source Data file.

changes arise from FRET between RNC$^{BDP}$ and SRP$^{TMR}$ upon their binding (Fig. 3b).

Equilibrium titrations based on this FRET assay showed that SRP$^{TMR}$ binds RNC(ss) and RNC(ssmt) with equilibrium dissociation constants ($K_{d,SRP}$) of 0.44 and 1.6 nM, respectively (Fig. 3c). The $K_{d,SRP}$ for RNC(ss) is comparable to previously measured values for RNC(pPL) reported by Flanagan et al.[14]. The $K_{d,SRP}$ for RNC(ssmt) is approximately 5-fold lower than that for RNC(globin)[14], likely due to the more hydrophobic sequence of ssmt compared to the globin nascent chain. Sub-micromolar concentrations of NAC reduced the FRET signal between SRP and RNC(ss), as well as RNC(ssmt) (Fig. 3d), suggesting that NAC regulates the affinity and/or the conformation of RNC-SRP binding. The ribosome binding deficient NACmt had negligible effects on the observed FRET signal unless added at high micromolar concentrations (Fig. 3d, gray and black), indicating that the ribosome interaction of NAC is necessary for its regulation of RNC-SRP binding.

To quantitatively measure the effects of NAC on the binding of SRP to RNC, we performed RNC-SRP FRET titrations in the presence of increasing NAC concentrations (Fig. 3e, f). The data were globally fit to the model in Fig. 3g, which describes the interactions of SRP and NAC with RNC using three parameters: the binding affinities of RNC for SRP ($K_{d,SRP}$) and NAC ($K_{d,NAC}$) and the coupling coefficient α that describes the allosteric effect of SRP and NAC on the RNC binding affinity of one another. An α value less than one indicates cooperative binding of NAC and SRP to the RNC, whereas an α value larger than one indicates anti-cooperative binding between NAC and SRP. The FRET titration data for both RNC(ss) and RNC(ssmt) over a wide range of NAC concentrations fit well to the anti-cooperative model (Fig. 3e-g). The $K_{d,SRP}$ values for both RNC(ss) and RNC(ssmt) obtained from the global fit are in good agreement with those from Fig. 3c (0.36 vs. 0.44 nM for RNC(ss) and 1.5 vs. 1.6 nM for RNC(ssmt)). These data also showed that NAC binds both RNC (ss) and RNC(ssmt) tightly, with $K_{d,NAC}$ values of 1.4 and 1.2 nM, respectively (Fig. 3g, right panel). The coupling coefficient α obtained from these data were 6.6 and 2.8 for RNC(ss) and RNC (ssmt) (Fig. 3g, right panel), indicating that SRP and NAC modestly weaken the binding of each other to the RNC.

We also globally fitted these data to the alternative model in which SRP and NAC competes with each other for RNC binding (Supplementary Fig. 4a–c). Even with the best-fit parameters, there were substantial deviations between the fit and experimental data. In addition, the apparent $K_{d,SRP}$ values at different NAC concentration, obtained from the individual titration curves, saturated at low NAC concentrations and were well matched by predictions from the anti-cooperative model (Supplementary Fig. 4d, e). In contrast, the apparent $K_{d,SRP}$ values would rise linearly with increasing NAC concentration in the competitive model, which yielded predictions that deviate from the experimental data by over an order of magnitude for both RNCs (Supplementary Fig. 4d, e). These analyses further support the anti-cooperative model and exclude the competitive binding model.

To independently test this model, we directly measured the binding affinity of NAC for RNC. We labeled NAC with Cy3B-maleimide at an engineered single cysteine (C57) in an unstructured N-terminal region of NACβ that mediates ribosome binding[31,32] (Supplementary Fig. 5a). The fluorescence intensity of RNC$^{BDP}$ was reduced ~30% in the presence of NAC$^{Cy3B}$ with a corresponding increase in Cy3B fluorescence (Fig. 4a and Supplementary Fig. 5b), and the fluorescence change could be competed away by a 5-fold excess of unlabeled NAC (Supplementary Fig. 5b), indicating FRET between RNC$^{BDP}$ and NAC$^{Cy3B}$ upon their binding. Equilibrium titrations using this assay showed that NAC$^{Cy3B}$ binds RNC(ss) and RNC(ssmt) with

$K_{d,NAC}$ values of 1.6 and 1.3 nM, respectively (Fig. 4c, d, black), similar to those obtained from global fits of the data in Fig. 3 (1.4 and 1.2 nM, respectively). Using unlabeled NAC as a competitor for NAC$^{Cy3B}$, we found that unlabeled NAC binds both RNCs with affinities within ~2-fold of NAC$^{Cy3B}$ (Fig. 4b), indicating that fluorescence labeling of NAC did not substantially perturb its RNC binding. Finally, Cy3B-labeled NACmt did not display significant FRET with the RNCs until NAC concentration was raised above 100 nM (Fig. 4c, d, gray), and unlabeled NACmt did not affect the FRET signal between RNC$^{BDP}$ and NAC$^{Cy3B}$ (Fig. 4b, gray and black), confirming that interaction with the ribosome is crucial for RNC-NAC binding.

Addition of SRP modestly shifted the RNC-NAC titration curves, raising the value of $K_{d,NAC}$ ~6-fold for RNC(ss) and <2-fold for RNC(ssmt) at saturating SRP concentrations (Fig. 4c, d). These effects are in close agreement with the coupling coefficient α determined in Fig. 3. Furthermore, we fitted the individual NAC titration curves to obtain the apparent $K_{d,NAC}$ values at each SRP concentration (Fig. 4e, f). The experimental SRP concentration dependences of $K_{d,NAC}$ were closely matched by predictions from the anti-cooperative model, but deviated by over an order of magnitude from the competitive model (Fig. 4e, f). These results provide independent support for the co-binding of SRP and NAC on RNC.

**Detection of the RNC-NAC-SRP ternary complex.** To directly detect the RNC-SRP-NAC ternary complex, we performed single-molecule (sm) colocalization experiments using total internal reflection fluorescence (TIRF) microscopy with alternating laser excitation (ALEX). We re-engineered the RNC constructs to replace the C-terminal 23 residues of the nascent chain with a mammalian translation stall sequence derived from Xbp1u[40] (Supplementary Fig. 6b) followed by a 500-nucleotide 3′-UTR, so that the RNCs contain a free mRNA 3′-end for biotinylation and coupling to microscope slides. The microscope slide surface incubated with biotinylated RNC had a high density of fluorescently labeled SRP whereas that with non-biotinylated RNC showed minimal fluorescent spots (Supplementary Fig. 6a), confirming the specificity of the immobilization.

Using this smTIRF microscopy setup, we tested for the colocalization of NAC$^{Cy3B}$ and SRP$^{Atto647n}$ to surface-immobilized RNCs. We recorded movies of immobilized ribosomal complexes and extracted the fluorescence time traces from diffraction-limited spots that displayed single step photobleaching or photoblinking. Even at NAC$^{Cy3B}$ and SRP$^{Atto647N}$ concentrations below 2 nM, we observed multiple colocalization events between NAC$^{Cy3B}$ and SRP$^{Atto647n}$ on surface-immobilized RNC(ss) and RNC(ssmt) (Fig. 5). Further, we detected FRET between NAC and SRP in some of the colocalization events, as indicated by the anti-correlation between the donor and acceptor emission channels during donor excitation (Fig. 5a, b). In no cases did we observed the dissociation of SRP coincident with the binding of NAC. As controls, we observed no colocalization between SRP and the ribosome binding-deficient NACmt, or in the absence of surface-immobilized RNC (Fig. 5h). These data provide direct evidence for the co-binding of SRP and NAC to the RNC and further indicate close proximity between NAC and the SRP54 NG-domain in the ternary complex.

Collectively, multiple independent measurements showed that SRP and NAC co-bind to RNCs and modestly weaken the binding affinity of one another. The predicted binding affinities of SRP for NAC-bound RNC(ss) and RNC(ssmt) (Fig. 3g, α•$K_{d,SRP}$) were 2.4 and 4.5 nM, respectively, and the predicted affinities of NAC for SRP-bound RNC(ss) and RNC(ssmt) (Fig. 3g, α•$K_{d,NAC}$) were 9.2 and 3.4 nM, respectively. Thus, SRP was fully bound by

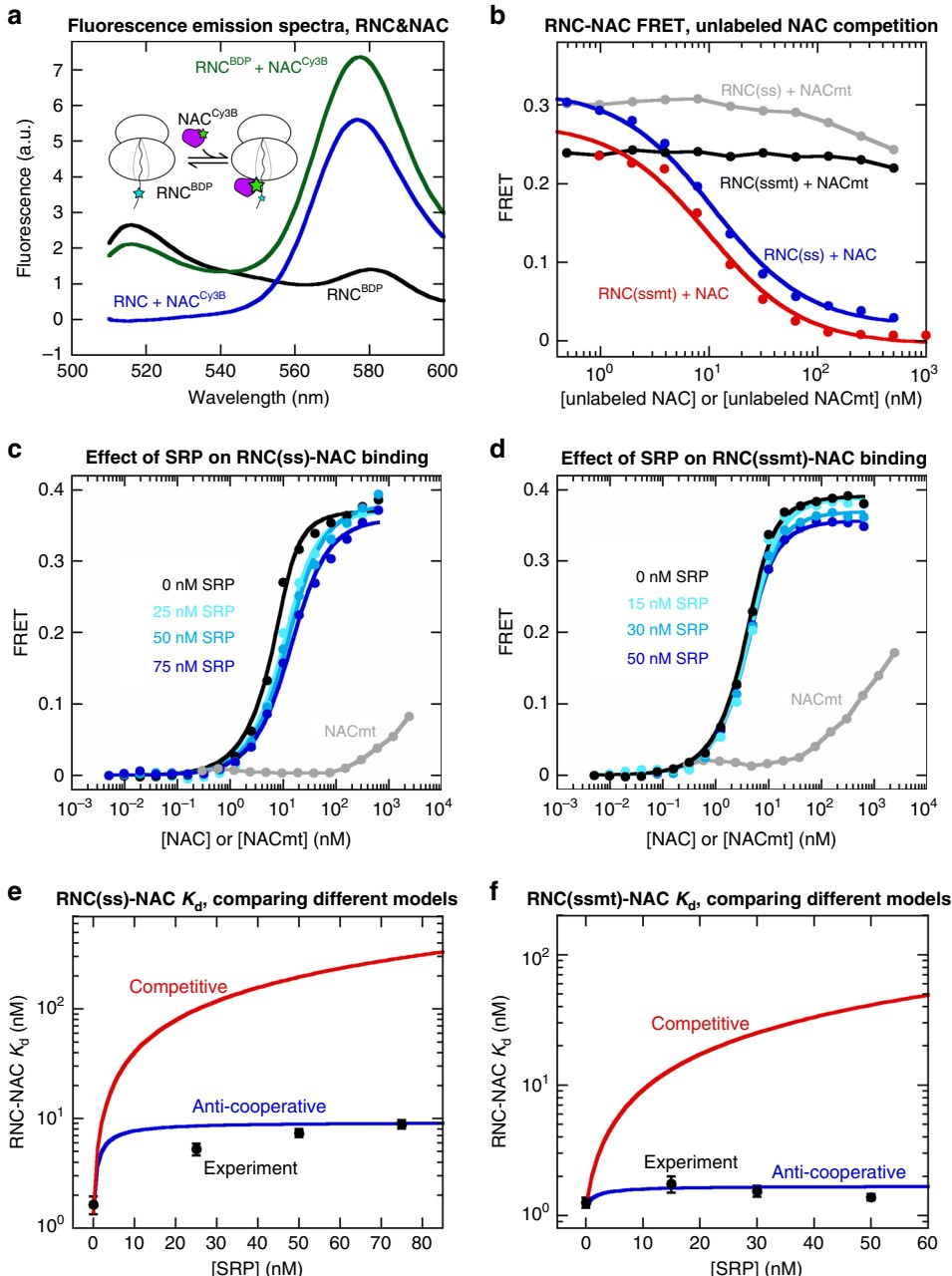

**Fig. 4 An RNC-NAC binding assay shows co-binding of SRP and NAC on the RNC. a** Inset: scheme of the FRET assay to measure RNC-NAC binding. RNC was labeled with BDP (cyan star) at the N-terminus of signal sequence, and NAC was labeled with Cy3B (green star) at residue 57 of NACβ. Figure: Fluorescence emission spectra showing FRET between labeled RNC and NAC. Where indicated, the reactions contained 1 nM RNC(ss)$^{BDP}$ or 15 nM unlabeled RNC(ss) and 5 nM NAC$^{Cy3B}$. The spectra were measured by excitation at 485 nm. **b** Competition assay to determine the RNC binding affinity of unlabeled NAC and NACmt. 1 nM RNC$^{BDP}$ and 20 nM NAC$^{Cy3B}$ were pre-incubated to form the RNC$^{BDP}$-NAC$^{Cy3B}$ complex, followed by addition of unlabeled NAC or NACmt to compete with NAC$^{Cy3B}$ for RNC binding. The data with NAC were fit to Eq. (6) in the Methods, which gave $K_d$ values of 0.55 and 0.50 nM for the binding of unlabeled NAC to RNC(ss) and RNC(ssmt), respectively. **c, d** Equilibrium titrations to measure the binding of NAC to RNC (ss) (**c**) and RNC(ssmt) (**d**) in the presence of increasing SRP concentrations. The reactions contained 5 nM RNC$^{BDP}$ and the indicated concentrations of NAC$^{Cy3B}$, unlabeled NAC and SRP. The data were fit to Eq. (5) to obtain apparent $K_{d,NAC}$ values at the individual SRP concentrations. **e, f** Comparison of the experimentally determined apparent $K_{d,NAC}$ values at different SRP concentrations (black squares) with predictions from the anti-cooperative model in Fig. 3g (blue line, simulated using Eq. (10)) and the competitive model in Supplementary Fig. 4a (red line, simulated using Eq. (11)). The simulations in **e** used $K_{d,SRP}$, $K_{d,NAC}$ and α values of 0.36 nM, 1.4 nM, and 6.6, respectively. The simulations in **f** used $K_{d,SRP}$, $K_{d,NAC}$ and α values of 1.5 nM, 1.2 nM, and 1.4, respectively. The experimental data were shown as mean ± SD, with $n = 3$ independent measurements on the same biological sample. Source data for **a**–**f** are provided in the Source Data file.

RNC•NAC in the SRP-SR association measurements in Fig. 2. Further, the smaller anti-cooperativity between SRP and NAC for binding RNC(ssmt) compared to RNC(ss) is contrary to the larger inhibitory effect of NAC on SRP-SR association with RNC(ssmt) than with RNC(ss). Thus, the observed effects of NAC on SRP-SR association (Fig. 2d–g) cannot be attributed to the exclusion of SRP from the RNC and must instead arise from NAC-induced allosteric regulation of SRP in a ternary RNC-SRP-NAC complex.

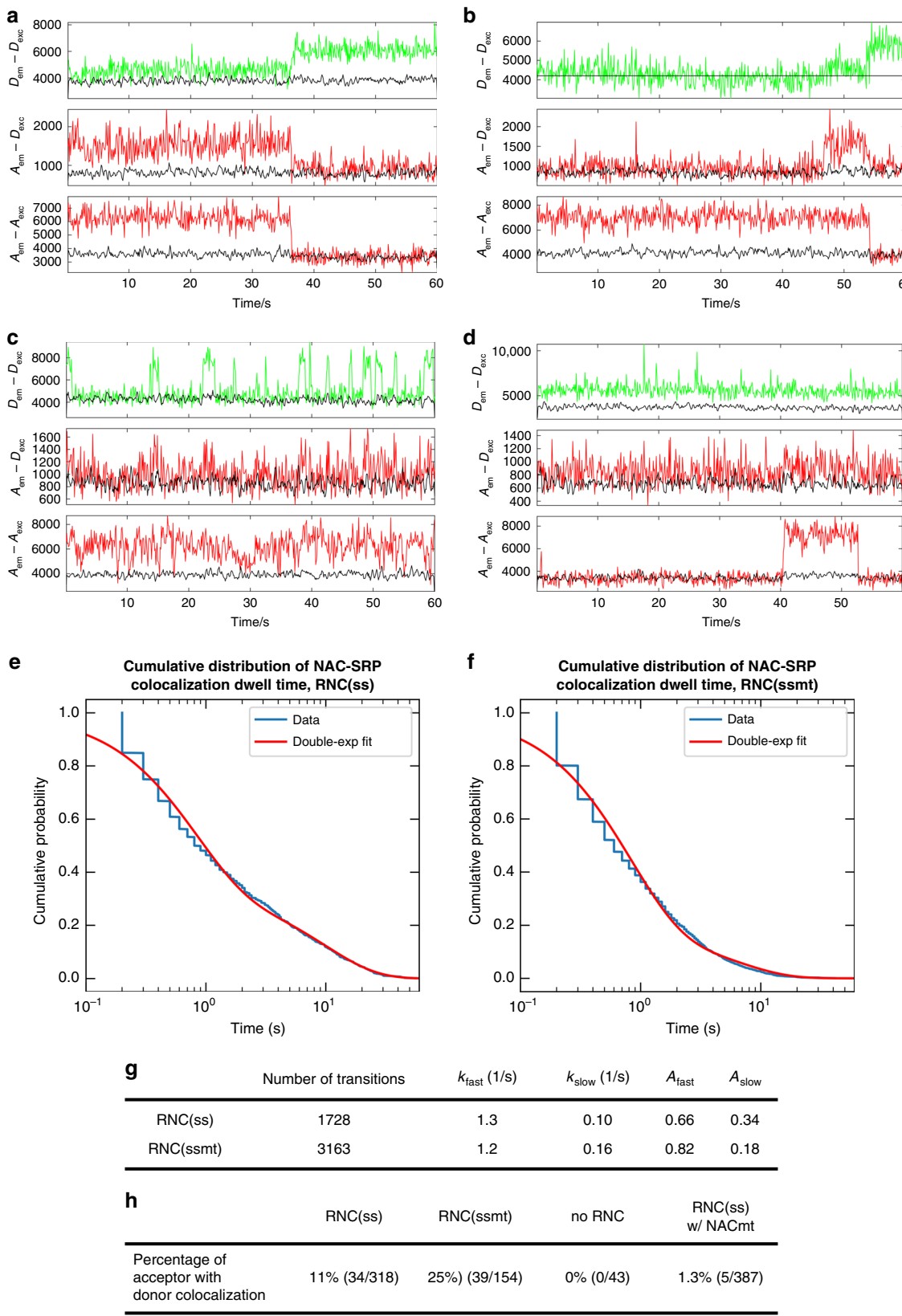

| g | Number of transitions | $k_{fast}$ (1/s) | $k_{slow}$ (1/s) | $A_{fast}$ | $A_{slow}$ |
|---|---|---|---|---|---|
| RNC(ss) | 1728 | 1.3 | 0.10 | 0.66 | 0.34 |
| RNC(ssmt) | 3163 | 1.2 | 0.16 | 0.82 | 0.18 |

| h | RNC(ss) | RNC(ssmt) | no RNC | RNC(ss) w/ NACmt |
|---|---|---|---|---|
| Percentage of acceptor with donor colocalization | 11% (34/318) | 25%) (39/154) | 0% (0/43) | 1.3% (5/387) |

**Signal sequence-dependent regulation of SRP conformation by NAC.** To directly observe NAC-induced conformational changes of SRP on the RNC, we used a FRET pair in SRP that reports on the proximity between the SRP54 NG-domain (labeled with Atto647n as the acceptor dye) and SRP19 (labeled with Atto550 as the donor dye) adjacent to the ribosome exit site[10] (Fig. 6a). Previous solution-based single-molecule (sm) FRET measurements using this dye pair showed that the ribosome and signal sequence sequentially drive SRP into a 'proximal' conformation characterized by high FRET between the dye pair. The FRET efficiency in this state is consistent with the cryoEM structure of the RNC·SRP complex, in which the SRP54-NG domain docks at ribosomal proteins uL23/uL29 near the

**Fig. 5 Single-molecule colocalization of NAC and SRP on surface immobilized RNCs. a–d** Representative traces of NAC-SRP colocalization on surface immobilized RNC(ss) (**a**, **b**) and RNC(ssmt) (**c**, **d**). Biotinylated quartz surface was coated with 1.5 nM purified monosome RNC(ss) or RNC(ssmt) with 3′ biotinylated mRNA. The sample chamber was then flushed with 2 nM NAC$^{Cy3B}$ and 1 nM SRP$^{Atto647n}$ (labeled at SRP54-S12C) in image buffer. Movies were recorded in ALEX mode for 60 s at a speed of 10 frames per second. Note the anti-correlation in $A_{em}$–$D_{ex}$ and $D_{em}$–$D_{ex}$ panels that indicates FRET between NAC$^{Cy3B}$ and SRP$^{Atto647n}$. **e–g** Dwell time analysis of the NAC binding events to RNC(ss) (**e**) and RNC(ssmt) (**f**). Single molecule fluorescence traces were fit to a Hidden Markov Model (HMM) to extract dwell times of NAC in the colocalized state. The cumulative distributions of dwell times were fitted to a double-exponential function Eq. (17) in the Method, and the fitted parameters are reported in (**g**). Number of transition is the total number of transitions observed for NAC binding to and dissociation from the RNC-SRP complex. **h** Summary of the frequency of observed SRP-NAC colocalization events under each condition, calculated from the ratio of the number of acceptors with colocalized donor over the total number of acceptors detected. No RNC indicates that 2 nM NAC$^{Cy3B}$ and 1 nM SRP$^{Atto647n}$ (labeled at 54-S12C) were incubated in image buffer on microscope slides without immobilized RNC. RNC(ss) w/ NACmt is the same as RNC(ss) except that 2 nM of NACmt instead of NAC was used.

exit tunnel and is in close proximity to SRP19[6]. The population of SRP in the high FRET state strongly correlates with the activation of SRP-SR association kinetics, indicating that this conformation of SRP is optimal for SR binding[10].

We carried out smFRET measurements of RNC-bound SRP using TIRF microscopy with alternating laser excitation (Fig. 6a). As SRPs were recruited to the TIRF illumination window via the surface-immobilized RNC, only RNC-bound SRP would be detected in this setup. We recorded movies of immobilized SRP-RNC and extracted the fluorescence time traces from diffraction-limited spots that (i) showed colocalized fluorescence signals from both the donor and acceptor dyes; and (ii) displayed single step photobleaching or photoblinking (Fig. 6b, first three rows). We pooled the FRET efficiency from the individual time frames across a large number of traces to construct smFRET histograms, which report on the conformational distribution of SRP (Fig. 6c, d). The smFRET histograms observed with the TIRF setup were comparable to those from previous solution-based smFRET measurements of SRP bound to both RNC(ss) and RNC(ssmt)[10] (Supplementary Fig. 6c, d). The histograms were best fit by the sum of three Gaussian distributions with low, medium and high FRET, as reported[10]. The distribution of SRP among the different FRET populations measured under the TIRF setup was also comparable to that from solution-based smFRET measurements (Supplementary Fig. 6e).

smFRET distributions of SRP in the presence and absence of saturating NAC (300 nM) were measured and compared for RNC(ss) and RNC(ssmt). The smFRET histogram of RNC(ss)-bound SRP was dominated by the high FRET population, which was reduced modestly in the presence of NAC, from 0.97 to 0.77, with a corresponding small increase in the low FRET population (Fig. 6c, e). In contrast, RNC(ssmt)-bound SRP showed a broad conformational distribution, sampling the low, medium and high FRET states with substantial probability (Fig. 6d, e), similar to previous observations with ribosome-bound SRP[10]. The addition of NAC nearly eliminated the high FRET population (from 0.31 to 0.050), while significantly increasing the medium-FRET population that characterizes an SRP conformation inactive in SR binding[10]. These changes in the conformational distribution of SRP correlated with the NAC-induced changes in SRP-SR association kinetics on both RNCs.

Thus, smFRET measurements demonstrated that NAC reduced the proximal conformation of SRP on the ribosome that is optimal for SR binding and provided additional support for the co-binding of SRP and NAC. The conformational regulation by NAC is selective for SRPs bound to signalless RNCs and largely accounts for the NAC-induced inhibition of SRP-SR association on suboptimal substrates.

**NAC suppresses the pre-emptive targeting of ribosomes with a short nascent chain.** To understand how NAC regulates SRP during ongoing protein synthesis, we further measured SRP-SR

association kinetics on RNC(ss) and RNC(ssmt) at different nascent chain lengths (Fig. 7a and Supplementary Fig. 7). Both in the presence and absence of NAC, SRP-SR association on RNC(ss) accelerated significantly as the nascent chain elongates from 35 to 45 amino acids (or 21 to 31 amino acids after the signal sequence), but remained largely invariant at shorter or longer nascent chain lengths (Fig. 7a, cyan and navy). As the nascent polypeptide exit tunnel accommodates ~35 amino acids[41], these results demonstrate the activation of SRP-SR interaction upon the emergence of a signal sequence from the ribosome tunnel exit (Fig. 7a, red shaded region). In contrast, SRP-SR association rates on RNC(ssmt) were independent of nascent chain length (Fig. 7a, red and brown). Further, the rates observed with RNC(ss) and RNC(ssmt) differ by less than two-fold before the nascent chain reaches 35 aa (Fig. 7a, cyan vs. red, and navy vs. brown). These results indicate that an unexposed signal sequence does not significantly activate SRP-SR association, and that the ribosome is dominant in governing SRP-SR assembly in the absence of an exposed functional signal sequence.

Importantly, NAC strongly inhibited SRP-SR association on RNC(ssmt) across all nascent chain lengths (Fig. 7a, red vs. brown). Further, NAC caused a 20-fold reduction in the rate of SRP-SR association on RNC(ss) before the nascent chain reaches 35 aa, indicating that NAC also delays the onset of targeting until the signal sequence emerges from the ribosome exit tunnel (Fig. 7a, cyan and navy). We also verified that SRP binds the 80S ribosome with high affinity ($K_{d,SRP}$ ~ 20 nM) and that NAC did not weaken the binding affinity of SRP to 80S (Supplementary Fig. 8). Thus, the observed effects of NAC on SRP-SR assembly rates with short-chain RNCs are unlikely to arise from weakened SRP binding to these RNCs. Together, these results demonstrate that NAC regulates SRP across a range of nascent chain lengths to suppress both the nonspecific targeting of signalless ribosomes and the pre-emptive targeting of ribosomes when the signal sequence is still buried inside the exit tunnel.

**Kinetic modeling emphasizes the role of NAC in the specificity of cotranslational protein targeting.** To quantitatively understand how the NAC-induced regulation of SRP observed in our reconstituted system impact this pathway under in vivo-like conditions, we constructed an analytical kinetic model for cotranslational protein targeting by SRP[42] (Fig. 7b). In this model, ongoing protein synthesis is described by an elongation rate constant ($k_{elongation}$). At each nascent chain length, the RNC could recruit SRP ($k_{on,SRP}$ and $k_{off,SRP}$) and activate it for SR binding ($k_{on,SR}$ and $k_{off,SR}$) to form the RNC-SRP-SR complex, followed by an irreversible step that commits the RNC for translocation ($k_{target}$). The rate and equilibrium constants for RNC-SRP and SRP-SR binding were either directly determined in this and recent works[10,43] or were estimated from experimental measurements (see Fig. 7d and details in Methods section). As described by Sharma et al.[44], solving the differential rate

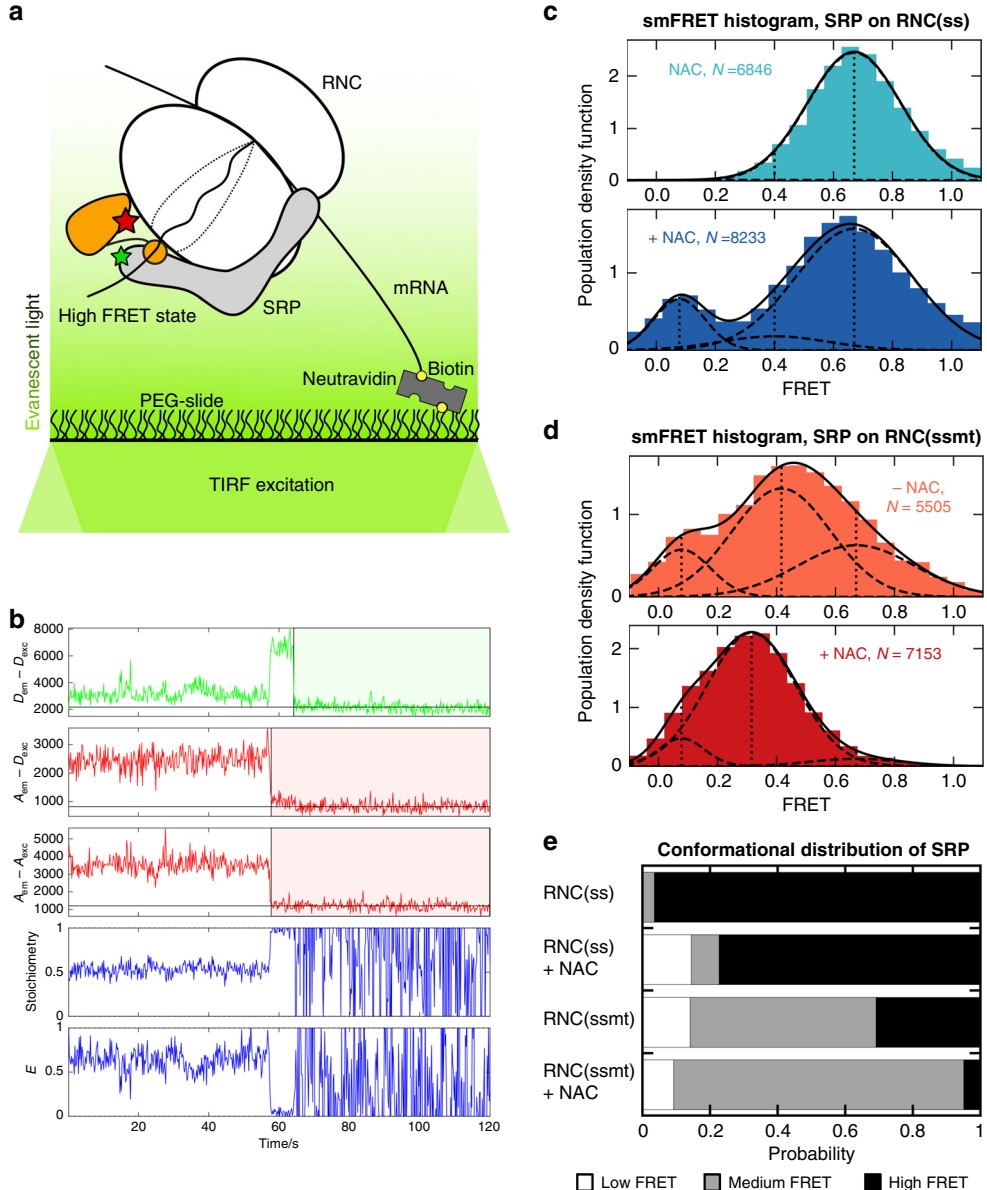

**Fig. 6 smFRET-TIRF microscopy detects NAC-induced conformational changes of SRP. a** Scheme of the smFRET-TIRF assay to monitor the global conformation of RNC-bound SRP. RNC was immobilized on the microscope slide via biotinylation of the mRNA 3'-end and recruits doubly labeled SRP to the TIRF illumination window. Green and red stars depict the donor and acceptor dyes on SRP, respectively, which gains high FRET in the proximal conformation as depicted. The RNC-SRP complex was pre-formed with 100 nM RNC and 10 nM SRP, diluted 20-fold in image buffer, and loaded onto the PEGylated quartz slide doped with neutravidin. The donor and the acceptor dyes were alternatively excited with 100 ms time intervals. **b** Representative single molecule fluorescence time traces. The first three traces show donor emission when exciting donor ($D_{em}$–$D_{exc}$), acceptor emission when exciting donor ($A_{em}$–$D_{exc}$), and acceptor emission when exciting acceptor ($A_{em}$–$A_{exc}$). The vertical lines indicate photobleaching events, and the shaded areas denote time intervals when the fluorophores were in the dark state. Anti-correlation between $D_{em}$–$D_{exc}$ and $A_{em}$–$D_{exc}$ is observed when the acceptor photobleaches at ~60 s, corroborating FRET between the dye pair. In each time frame, FRET efficiency ($E$) and stoichiometry ($S$) were calculated using Eqs. (12) and (13) in the Methods section, respectively. The data during the dark states of either fluorophore were discarded and not included in the smFRET histogram. **c, d** smFRET histograms of SRP bound to RNC(ss) (**c**) and RNC(ssmt) (**d**) in the absence and presence of NAC (solid lines). N is the number of frames collected. Each histogram was fit to the sum of three-Gaussian distributions representing low, medium, and high-FRET populations (solid lines) using Eq. (16), with the dashed lines indicating individual Gaussian distributions, and the vertical dotted lines indicating the mean FRET value of each population. **e** Summary of the effect of NAC on the conformational distribution of SRP. The relative population of SRP in the low, medium and high FRET states were plotted as cumulative bar graphs for SRP bound to RNC(ss) and RNC(ssmt) in the absence and presence of NAC. Source data for **e** are provided in the Source Data file.

equations defined in the model allows the fraction of successfully targeted RNCs to be determined at each nascent chain length, generating progression curves for SRP-mediated protein targeting during ongoing protein synthesis.

The model showed similar progression curves for the cotranslational targeting of RNC(ss) and RNC(ssmt) in the absence of NAC: 50% of the RNC would be targeted when the nascent chain is <60 amino acids long, and targeting is close to

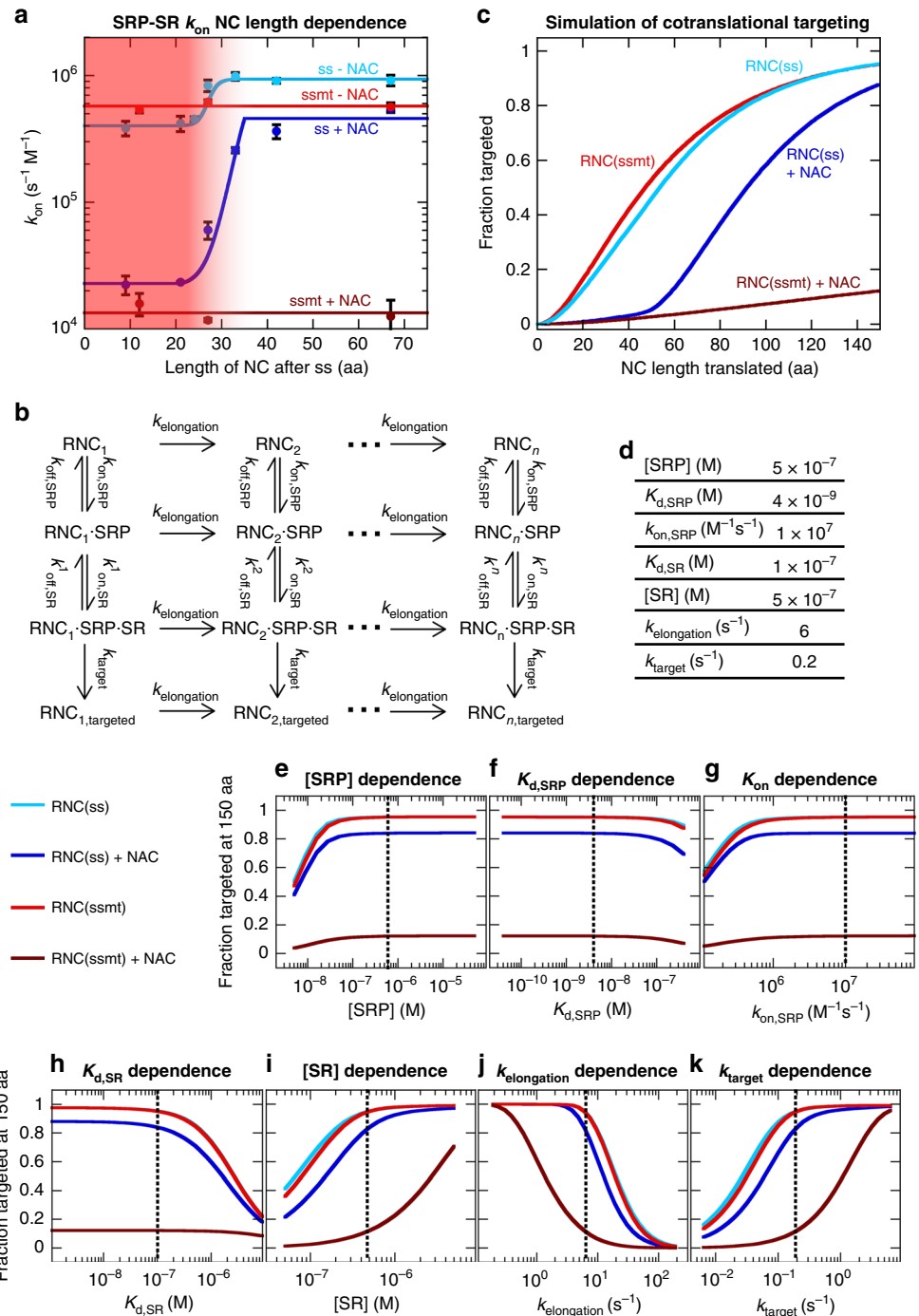

**Fig. 7 A computational model for the SRP pathway emphasizes the role of NAC in maintaining targeting specificity. a** NC length dependence of SRP-SR association rate constants. All measurements used the same concentrations of RNC, SRP and NAC as in Fig. 2g. The red shaded area indicates NC lengths at which the signal sequence is partially or completely buried, assuming that the ribosome exit tunnel accommodates ~35 amino acids. The data were fit to Eq. (18) in the Methods section (solid lines) for RNC(ss) and to a constant $k_{on}$ value for RNC(ssmt). All experimental data are shown as mean ± SD, with $n = 3$–5 independent measurements on the same biological sample. **b** Computational model for co-translational protein targeting by SRP and SR. **c**, **d** Modeled progression curves for SRP-dependent protein targeting during ongoing protein synthesis with and without NAC present (**c**). The model in **b** was calculated as described in the Methods section, assuming that the signal sequence is located within the N-terminal 14 amino acids of the nascent protein and that the exit tunnel accommodates 35 amino acids. Values of $k_{on,SR}$ were from (**a**) (solid lines). The other parameters used for the modeling are summarized in **d** and detailed in the Methods section. **e–k** Sensitivity of the modeled targeting efficiency to perturbations of the individual parameters in the model. The fraction of successfully targeted RNCs at a nascent chain length of 150 aa was determined as in **c**, except that each of the parameters listed in **d** was varied by 1–2 orders of magnitude from the estimated values (dotted lines). Source data for **a**, **c**, and **e–k** are provided in the Source Data file.

completion at >100 amino acids (Fig. 7c, cyan and red). The early targeting is due to rapid SRP-SR association on short-chain RNCs in the absence of NAC, and the lack of targeting specificity is due to the small difference between the SRP-SR association rates on RNC(ss) versus RNC(ssmt) without NAC present. The presence of NAC introduced two major changes (Fig. 7c, navy and brown). First, 'pre-emptive' targeting before the nascent chain reaches 45 amino acids, when the signal sequence is partially or completely buried inside the ribosome exit tunnel, was reduced to negligible levels. Secondly, NAC significantly enhanced the specificity of SRP-dependent targeting upon emergence of the signal sequence from the ribosome, reducing the incorrect targeting of RNC(ssmt) to <20%. In contrast, even in the presence of NAC, targeting of RNC(ss) proceeds efficiently upon exposure of the signal sequence from the ribosome tunnel exit, reaching ~90% completion when the nascent chain is 150 amino acids long.

To understand the contribution of various factors to cotranslational protein targeting, we calculated the fraction of successfully targeted RNCs when the nascent chain reaches 150 aa and tested the sensitivity of the modeling results to variations in the individual parameters in the model (Fig. 7e–k). We found that the predicted targeting efficiencies remained largely the same when the concentration of SRP or the kinetics and affinity of RNC-SRP binding ($k_{on,SRP}$ and $K_{d,SRP}$) were varied by 1–2 orders of magnitude within their estimated limits (Fig. 7e–g). The modeling result is also insensitive to <100-fold changes in RNC-SRP $K_d$ in response to nascent chain length[14,45] (Supplementary Fig. 9). This strongly suggests that the in vivo SRP concentration is saturating with respect to its RNC binding affinity and that SRP-RNC binding is not rate-limiting for the overall targeting reaction. Analogously, the modeled targeting efficiency is robust to variations in the affinity of the SRP•SR complex ($K_{d,SR}$) on RNCs, reflecting the fact that the in vivo concentration of SR is saturating with respect to the SRP-SR binding affinity on RNCs (Fig. 7h). On the other hand, the targeting efficiencies are sensitive to variations in the rate of translation elongation ($k_{elongation}$), the commitment of the targeting complex to translocation ($k_{target}$), and the concentration of SR (Fig. 7i–k), suggesting potential cellular and molecular mechanisms for regulation of the SRP pathway.

In summary, the kinetic measurements in this work allow construction of an analytical mathematical model to describe cotranslational protein targeting by SRP. Our model demonstrates that the regulatory effects of NAC on SRP, observed in the biochemical and single molecule measurements, play essential roles in maintaining the fidelity of protein targeting under in vivo-like conditions.

## Discussion

Accumulating data show that protein biogenesis begins when the nascent polypeptide emerges from the ribosome tunnel exit, where multiple RPBs can bind in the vicinity and compete for access to the nascent chain. Engagement with these RPBs commits the nascent protein to distinct protein biogenesis pathways, and mistakes in these early events can lead to devastating consequences for the cell[46–49]. The molecular crowding at the tunnel exit creates opportunities for coordination and regulation, both spatially and temporally, between different protein biogenesis pathways (Fig. 1a). Spatially, RPBs can compete for the same binding site, or co-bind on the RNC to regulate one another. Temporally, the ribosome association, conformation, and activity of RPBs could be modulated during elongation of the nascent chain, generating time windows for the action of individual RPBs that can be regulated by translation elongation rates[50]. How multiple RPBs coordinate with one another in space and time at

the ribosome exit site and how this coordination impacts biological function remain unanswered questions. Our work here begins to address these questions by studying two major eukaryotic cotranslational protein biogenesis machineries, SRP and NAC, as a model system.

Deciphering the molecular interplay of RPBs relies critically on high resolution methods that can interrogate the interaction and conformation of RPBs on the ribosome. In this work, we adopted bioorthogonal amber suppression mediated by the engineered *Mm* PyltRNA/RS pair to the RRL in vitro translation system, which provides a facile and efficient method for site-specific incorporation of fluorescent probes into nascent proteins on mammalian ribosomes. This enables quantitative measurements of the energetics and kinetics of the interaction of RNC with SRP and NAC, and these assays could be readily extended to other RPBs. Compared to previously described systems or commercial fluorescent tRNAs, which use tRNAs chemically charged with non-natural amino acids[14,51], our method provides higher efficiency, robustness, ease of execution, and flexibility in choosing the labeling sites or motifs.

Previous crosslinking[27–29] and structural[6,23] works suggested a competitive model in which NAC and SRP exclude each other from binding to the same ribosome. The quantitative RNC-RPB binding measurements in this work provided conclusive evidence against this model. Instead, we found that both SRP and NAC bind with subnanomolar to low-nanomolar affinity to ribosomes with or without a signal sequence. Further, they co-bind on the same RNC with modest anti-cooperativity, leading to effective SRP binding affinities of $K_d \leq 5$ nM for both RNC(ss) and RNC(ssmt) even with NAC bound on the same ribosome. These observations strongly suggest that most RNCs with or without a functional signal sequence will be bound by SRP at in vivo concentrations (~500 nM)[16]. Analogously, the surprisingly high affinity of NAC for both RNC(ss) and RNC(ssmt) and the near-stoichiometry concentrations of NAC[16,21] relative to the ribosome in vivo strongly suggest that all cytosolic ribosomes in the cell are likely bound by NAC unless physically blocked by other RPBs. The universal, tight binding of NAC near the ribosome tunnel exit suggests an important role of NAC in coordinating co-translational processes.

While previous studies focused on the inhibition of RNC-SRP or RNC-Sec61p binding by NAC, our analyses show that NAC regulates cotranslational protein targeting primarily by reshaping the conformational landscape of SRP to modulate SR recruitment rates. This is directly demonstrated by smFRET probes that report on the proximity between the SRP54 NG-domain to the ribosome tunnel exit, which showed that NAC significantly reduced the proximal conformation of SRP bound to RNC(ssmt) (Fig. 6). In addition, NAC substantially reduced the FRET efficiency between the nascent chain and SRP54-NG in the RNC(ssmt)-SRP complex (Fig. 3d). Both observations suggest displacement of SRP54-NG from its original docking site and are consistent with the reduced crosslink between SRP54 and signalless nascent chains upon the addition of NAC[27,28]. As both SRP54-NG and NAC dock near ribosomal protein uL23 at the exit tunnel, these results are most simply explained by a model in which NAC binding displaces SRP-NG from uL23 while the remainder of SRP remains bound to the ribosome. This demonstrates the conformational flexibility and adaptation of SRP in the crowded macromolecular environment near the tunnel exit.

Importantly, the NAC-induced rearrangements in SRP is more pronounced for RNC(ssmt) than for RNC(ss). Since the proximal conformation, in which SRP54-NG docks at uL23, is the active state for assembly with SR, the NAC-induced loss of this conformation provides a molecular model to explain the selective reduction in SRP-SR association kinetics on RNCs with no or weak signal sequences. This enhances the discrimination between ribosomes with and without a functional signal sequence during

SRP-SR assembly, thereby increasing the specificity of cotranslational protein targeting. We note that in the *E. coli* SRP pathway, efficient association between SRP and SR strictly requires a signal sequence on the RNC[18,52]. In contrast, assembly of the mammalian SRP•SR complex is strongly activated by signalless ribosomes[10], and we showed here that RPBs such as NAC are required to suppress this nonspecific SRP-SR association. Quantitative kinetic modeling further demonstrates that the effect of NAC on SRP-SR association is essential for maintaining the specificity of protein targeting (Fig. 7). It appears that, while the bacterial SRP and SR comprise a self-sufficient system to generate high fidelity protein targeting, substrate selection by the mammalian SRP pathway is strongly influenced by its associated macromolecular environment at the ribosome exit site.

Multiple recent studies suggested that SRP could associate with RNCs before the signal sequence emerges from the ribosome exit tunnel[6,17,53,54]. While these observations are intriguing, it was unclear whether these 'pre-emptive' binding events lead to targeting of the RNCs. Our data strongly suggest that, although the mammalian SRP can associate with short-chain RNCs, NAC plays a major role in delaying the onset of targeting. We found that SRP-SR association was fast on short-chain RNCs with or without a buried signal sequence in the absence of NAC, and that both were inhibited 10-fold to 20-fold by NAC. In computational modeling, these effects translate into a strong inhibition of the pre-emptive targeting of RNCs at nascent chain lengths below 50 aa in the presence of NAC (Fig. 7c). This may explain why RNCs with a buried signal sequence were found to associate with SRP but were not targeted to the ER in ribosome profiling experiments[17,55]. Thus, NAC also prevents RNCs from committing to ER targeting before the emergence of a signal sequence.

The quantitative measurements in this work provided sufficient information to construct a computational model for cotranslational protein targeting, which further allows us to test the robustness of the SRP pathway. For example, the pathway can tolerate perturbations of up to two orders of magnitude in SRP concentration, RNC-SRP binding affinity and/or kinetics. This can be attributed to the relatively high concentration of SRP in vivo compared to the $K_d$ of RNC-SRP binding, and emphasizes that the specificity of the SRP pathway cannot be maintained solely by differences in the binding affinity of SRP to different RNCs. The insensitivity of targeting efficiency to changes in SRP concentration or $K_{d,SRP}$ values in our model is also consistent with the observation that extensive reductions in SRP levels (>10-fold) are needed to observe targeting defects in mammalian cells[56]. Similarly, the efficiency and specificity of targeting are robust to changes in the affinity of the SRP•SR complex below a $K_{d,SR}$ value of 100 nM. We previously showed that the ribosome, rather than signal sequence, is responsible for stabilizing the SRP•SR complex, bringing the $K_{d,SR}$ value from >1 μM for free SRP to 40–80 nM for SRPs bound to the ribosome or RNC[10]. This and the in vivo SR concentration (~500 nM) imply that SRP-SR complex formation becomes thermodynamically favorable as soon as SRP is ribosome-associated, and the specificity of this process will likely arise from kinetic, rather than thermodynamic factors.

Computational modeling also identified potential mechanisms for regulation of the SRP pathway. For example, reducing SR concentration below its measured in vivo abundance impairs the targeting of signal sequence-containing RNCs, whereas higher SR concentrations would significantly increase the targeting of RNCs with suboptimal signal sequences. Intriguingly, glucose-induced stimulation of insulin secretion in pancreatic beta cells is associated with a 20-fold upregulation in SR abundance[57], which might be an example of a physiological adaptation based on this principle. The rate of translation elongation ($k_{elongation}$) is another important regulatory parameter (Fig. 7j), as variations in $k_{elongation}$ alters the time window available for SRP to target the nascent chain. Slower translation elongation is predicted to relax the specificity of SRP and enable the targeting of otherwise SRP-independent substrates (Fig. 7j); this has been observed in the SRP-mediated ER localization of the XBP1u mRNA, which requires a translation stall sequence in XBP1u[58]. Intriguingly, mammalian SRP contains an Alu-domain that has been shown to reduce translation elongation rate in vitro[59,60]. It was also suggested that selective codon usage to slow down translation elongation could occur when a signal sequence emerges from the ribosome tunnel exit to enhance targeting[61]. Whether these phenomena occur in vivo and how they contribute to protein targeting remain to be investigated. Finally, variations in $k_{target}$, the rate at which the RNC•SRP•SR complex engages the Sec61p translocon and commits to translocation, could substantially impact targeting (Fig. 7k). Notably, Sec61p also harbors a signal sequence/TMD binding site and could reject signalless RNCs[62–65], which provides an additional mechanism to enhance specificity beyond our current modeling results. Intriguingly, the measured or estimated values of $k_{target}$, $k_{elongation}$, and SR concentration are all at the optima in the tradeoff between targeting efficiency and specificity (Fig. 7i–k), suggesting that many parameters in the SRP pathway have adapted to the in vivo translation rates to balance between the two parameters. This observation also supports the notion that our computational model provides a reasonable framework to understand SRP-dependent cotranslational protein targeting in vivo.

In summary, our work provides a molecular model for how a major eukaryotic cotranslational chaperone, NAC, regulates the timing and specificity of cotranslational protein targeting via the SRP pathway. Without NAC, RNC-SRP interactions are dominated by the ribosome. Even RNCs with a weak signal sequences or a short nascent chain can induce a substantial population of SRP into the proximal conformation that is activated for rapid assembly with SR, leading to leaky and nonspecific delivery of translating ribosomes to the ER surface (Fig. 8, left panel). The presence of NAC at the ribosome exit site does not exclude SRP binding on the same RNCs but likely displaces SRP54-NG from its docking site near uL23 and selectively eliminates the proximal conformation for SRPs bound to RNCs with a weak signal sequence or a short nascent chain. This conformational change inhibits the SR recruitment and ER targeting of ribosomes without an exposed signal sequence (Fig. 8, right panel). As shown here and previously for prokaryotic SRP[52,66] and N-terminal methionine excision enzymes[42], the balance between efficiency and specificity of a cotranslational protein biogenesis pathway can be significantly reshaped by translation elongation and spatiotemporal coordination with other RPBs at the ribosome exit site. Our work provides a valuable conceptual framework, as well as experimental tools, to investigate this coordination on the mammalian ribosome at energetic and molecular detail, which can be readily extended to other cotranslational protein biogenesis pathways.

## Methods

**Plasmid construction**. Plasmids for expression of SRP proteins (SRP19, SRP9/14, SRP68/72 and SRP54), SR (SRα and SRβΔTM), and for in vitro transcription of 7SL RNA were described in Lee et al.[10]. Commercial cDNA clones of human NACα and NACβ (OriGene) were subcloned into pET15b using Gibson cloning to generate a bicistronic construct pET15b-NACα-NACβ for co-expression in *E. coli*. To enable metal affinity purification, the DNA sequence encoding the 6×His tag and PreScission protease site (MGSSHHHHHHSSGLEVLFQ/GPSG, / denotes the cleavage site) was inserted at the N-terminus of NACα using QuickChange mutagenesis (Stratagene). The ribosome binding mutant of NACβ ([27]RRK[29] to [27]AAA[29]) was generated using the QuickChange mutagenesis protocol (Stratagene). Plasmids encoding *Mm*PylRS and *Mm*PyltRNA were gifts from Dr. Jason

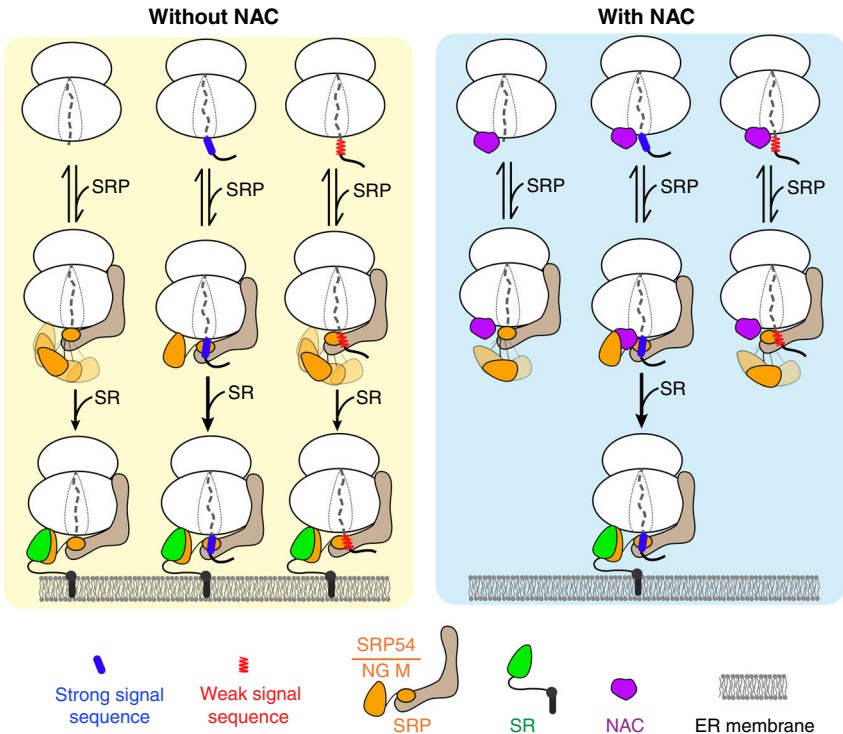

**Fig. 8 Model for how NAC allosterically regulates SRP to ensure the specificity of cotranslational protein targeting.** Left panel: Without NAC, the ribosome dominates the interactions of SRP, and even RNCs with a weak signal sequence (red zigzag line) or with a short nascent chain can induce SRP to dynamically sample the proximal conformation that rapidly associates with SR at the ER. Right panel: NAC (purple) co-binds with SRP to a variety of RNCs and selectively drives SRPs into inactive conformations on RNCs with a weak signal sequence or a short nascent chain, thus inhibiting the ER localization of ribosomes without an exposed and functional signal sequence.

Chin[38]. The *Mm*PylRS coding sequence was subcloned into pET15b to generate pET15b-*Mm*PylRS for expression in *E. coli* using Gibson cloning. The *Mm*PyltRNA coding sequence was subcloned behind the T7 promoter in a pUC19 plasmid using Gibson cloning to generate pUC19-*Mm*PyltRNA for in vitro transcription. For in vitro transcription of mRNAs, DNA encoding the encephalomyocarditis virus (EMCV) internal ribosome entry site (IRES)[67,68] (a gift from Dr. David Baltimore) was inserted using Gibson cloning upstream of the coding sequences for the specified nascent chains (ss, ssmt, and ssmt2) in a T7 promoter-containing pUC19 plasmid described by Lee et al.[10] to generate pUC19-IRES-NC. Amber codons were inserted 1-residue upstream of the signal sequence using QuickChange mutagenesis. For in vitro transcription of mRNA used in the translocation assay, plasmid pSPBP4 containing the pPL coding sequence and the 5′ and 3′-UTR of bovine pPL behind an SP6 promoter was used[69]. The pPL signal sequence was replaced with the indicated signal sequences (ss, ssmt, and ssmt2) to generate pSPBP4-pPL. To generate RNCs that contain the mRNA 3′UTR, the DNA sequence encoding an engineered strong translation stall sequence derived from Xbp1u[40] (YQPPFLCQWGRHQCAWKPLMN) replaces the coding sequence for the C-terminal 26 amino acids of the nascent chain using Gibson cloning. Primers used in this study are summarized in Supplementary Table 1.

**Protein expression and purification**
SRP and SR. Proteins were prepared essentially the same way as previously described[10]. SRP19 was expressed in Rosetta pLysS cells and purified with Ni-NTA and SP-Sepharose. SRP9 and SRP14 were expressed separately in BL21(DE3) pLysS cells, the lysates were mixed, and the SRP9/14 complex was purified with Heparin-Sepharose and MonoS cation exchange chromatography. SRP68 and SRP72 were co-expressed in yeast and purified with Ni-Sepharose and MonoS chromatography. SRP54 was expressed in Rosetta pLysS cells and purified with Ni-Sepharose and MonoS. The SRα wildtype or the GTP hydrolysis mutant (R458A) and SRβΔTM were co-expressed in BL21(DE3) and purified with Ni-Sepharose, MonoS cation exchange, and Superdex 200 size exclusion chromatography.

NAC. NACα and NACβ were coexpressed in BL21(DE3). Cells were grown to O. D. ~0.6 at 37 °C, induced with 1 mM IPTG, and temperature was lowered to 18 °C to allow expression overnight. Harvested cells were resuspended in Lysis Buffer (50 mM KHEPES pH 7.5, 1 M NaCl, 6 mM βME, 1 mM AEBSF, 10% glycerol and Protease Inhibitor cocktail (GoldBio)) and lysed by sonication. Clarified cell lysate was incubated with Ni-Sepharose (GE Healthcare) equilibrated with lysis buffer at 1.5 mL/L cell for 1 h at 4 °C. The resin was washed with 20 CV of Ni-Wash Buffer (50 mM KHEPES pH 7.5, 1 M NaCl, 6 mM βME, 45 mM imidazole, 10% glycerol). Washed resin was incubated with 1 CV of Elution Buffer (50 mM KHEPES pH 7.5,

150 mM KOAc, 6 mM βME, 10 mg/L PreScission enzyme, 10% glycerol) at 4 °C overnight. Eluted protein was purified over a MonoQ column using a gradient of 100–500 mM NaCl in MonoQ buffer (50 mM KHEPES, pH 7.5, 2 mM DTT, 2 mM EDTA, 10% glycerol). Protein containing fractions tested by SDS-PAGE were pooled and concentrated to ~200 μM using $\varepsilon_{280} = 4980\ cm^{-1}M^{-1}$, supplemented with glycerol to a final concentration of 20%, and stored at –80 °C.

*Mm*PylRS. BL21(DE3) cells harboring pET15b-*Mm*PylRS were grown to O.D. ~0.6 at 37 °C, induced with 1 mM IPTG, followed by overnight expression at 18 °C. Harvested cells were resuspended in Lysis Buffer and lysed by sonication. Clarified lysate was incubated with Ni-Sepharose equilibrated with Lysis Buffer at 1.5 mL/L cell for 1 h at 4 °C. The resin was washed with 20 CV of Wash buffer. The protein was eluted using 10 CV of Ni-Elution buffer (50 mM KHEPES pH 7.5, 1 M NaCl, 6 mM βME, 500 mM imidazole, 10% glycerol). Protein containing fractions tested by SDS-PAGE were pooled, diluted with 4 volumes of MonoS Buffer (50 mM KHEPES, pH 7.5, 2 mM DTT, 10% glycerol), and purified over a MonoS cation-exchange column using a gradient of 100–600 mM NaCl in MonoS buffer. Protein-containing fractions were pooled and concentrated to ~100 μM using $\varepsilon_{280} = 30000\ cm^{-1}M^{-1}$, supplemented with glycerol to a final concentration of 20%, and stored at –80 °C.

**In vitro transcription and purification of RNA**
SRP 7SL RNA. The in vitro transcription and purification of a circularly permutated variant of 7SL were carried out as described[10]. Briefly, linearized plasmid containing 7SL RNA sequence was transcribed with T7 polymerase. Transcribed RNA was acid phenol extracted and purified over a denaturing polyacrylamide gel (100 mM Tris, 89 mM Boric Acid, 1.3 mM EDTA, 7 M Urea, and 10% acrylamide (29:1)). RNA extracted from the gel was dialyzed in 20 mM Tris-HCl (pH 7.5), flash frozen in liquid nitrogen, and stored at –80 °C.

*Mm*PyltRNA. Template for in vitro transcription was PCR amplified from pUC19-*Mm*PyltRNA and in vitro transcribed by T7 polymerase using the Megascript protocol (Ambion). Transcribed RNA was purified by acid phenol-chloroform extraction and ethanol precipitation. The pellet was resuspended in 50 mM KHEPES, pH 7.5 at 24 mg/mL as measured by $A_{260}$, and stored at –80 °C.

mRNA for in vitro translation. Templates for in vitro transcription were PCR amplified from pUC19-IRES-NC and pSPBP4-pPL and in vitro transcribed by T7 polymerase or SP6 polymerase, respectively, following the Megascript protocol. Transcribed mRNA was precipitated with 3 M LiCl. The pellet was resuspended in 50 mM KHEPES, pH 7.5 to 3 mg/mL as measured by $A_{260}$ and stored at –80 °C. To produce 3′-biotinylated mRNA for TIRF experiments, purified mRNA was ligated with a 5′-monophosphorylated 3′-biotinylated oligo (IDT) with T4 RNA ligase

(NEB). The ligated mRNA was purified using RNeasy Mini kit (Qiagen) following the manufacturer's protocol and stored at –80 °C.

**SRP assembly**. SRP was assembled as described[10]. In brief, refolded 7SL SRP RNA was sequentially incubated with SRP19, SRP68/72, SRP9/14, and SRP54 at 37 °C. Assembled holo-SRP was purified using DEAE-Sephadex (Sigma) anion exchange column. Elution fractions corresponding to fully assembled SRP were identified by $A_{260}$ measurements, pooled, and stored at –80 °C.

**RNC preparation**. The RRL in vitro translation mix was prepared similarly to established protocol[70], except that nuclease digestion of RRL (Green Hectares) was omitted. RNC was synthesized by translating mRNA in the RRL translation mix for 30 min at 32 °C[6] and purified as follows[10]. Translation reaction was layered on an equal volume of High Salt Sucrose Cushion (50 mM KHEPES pH 7.5, 1 M KOAc, 15 mM Mg(OAc)$_2$, 0.5 M Sucrose, 0.1% Triton, 2 mM DTT), and ribosome was pelleted by ultracentrifugation (100k rpm for 1 h at 4 °C in a TLA100.3 rotor). The ribosome pellet was resuspended in RNC Buffer (50 mM KHEPES pH 7.5, 150 mM KOAc, 2 mM Mg(OAc)$_2$) to ~1 μM and incubated with Anti-DYKDDDK magnetic agarose (Pierce; 0.03 volume of translation reaction) pre-equilibrated in RNC buffer at 4 °C for 1 h with constant rotation. The agarose beads were collected by a magnet and washed sequentially with 10 bead volumes of RNC Buffer with 300 mM KOAc, RNC Buffer with 0.1% Triton, and RNC Buffer. RNC was eluted by incubation in RNC Buffer with 1.5 mg/mL 3× FLAG peptide at 4 °C for 30 min with constant rotation. The eluted RNC was layered onto a 4.8 mL sucrose gradient (50 mM KHEPES pH 7.5, 500 mM KOAc, 10 mM Mg(OAc)$_2$, 10–30% Sucrose, 2 mM DTT) and ultracentrifuged at 50k rpm for 1.5 h at 4 °C in a SW55 rotor. Fractions corresponding to monosome were pooled, and RNC was pelleted by ultracentrifugation at 100k rpm for 1 h at 4 °C in a TLA100.3 rotor. The RNC pellet was resuspended in Assay buffer (50 mM KHEPES, pH 7.5, 150 mM KOAc, 5 mM Mg(OAc)$_2$, 0.04% NIKKOL, 2 mM DTT) to ~2 μM and stored at –80 °C.

**Fluorescence labeling**
SRP54-Cy3B, SRP54-Atto647n, SRP54-TMR, SRP19-Atto550, NAC-Cy3B, and NAC-TMR. Cyslite SRP54 harboring an engineered cysteine (C12 for Atto647n and TMR, or C47 for Cy3B)[10], cysless SRP19 with an engineered single cysteine (C64)[10], and NAC with an engineered single cysteine (C57; this work) were purified as the wildtype protein. The proteins were dialyzed into Labeling Buffer (50 mM KHEPES pH 7.5, 300 mM KOAc, 2 mM TCEP, 10% glycerol) and labeled with an 8-fold molar excess of the indicated maleimide-conjugated dyes at 25 °C for 2 h. The labeled proteins were purified from free dye by G25 size exclusion column in Storage Buffer (50 mM KHEPES, pH 7.5, 300 mM KOAc, 2 mM DTT, 20% glycerol) and stored at –80 °C.

SR-Atto647n. As previously described[10,71], purified SR with a C-terminal sortase tag was labeled with Atto647n by incubation with purified sortase-A (without a 6× His tag) and dye-conjugated peptide, GGGC-Atto647n, at a molar ratio of 1:4:8 at 25 °C for 4 h in Sortase Buffer (50 mM KHEPES, pH 7.5, 150 mM NaCl, 10 mM CaCl$_2$, 10% glycerol, 2 mM DTT, 0.02% NIKKOL). Labeled SR was purified by Ni-Sepharose, exchanged into Storage Buffer, and stored at –80 °C.

RNC-BDP. RNC was generated by in vitro translation of mRNA with an amber codon in RRL translation mix containing 1 μM *Mm*PylRS, 10 mg/L *Mm*PyltRNA, and 100 μM axial-trans-cyclooct-2-en-L-Lysine (TCOK) (SiChem) for 30 min at 32 °C. Translation reactions were layered on an equal volume of high salt sucrose cushion, and ribosome was pelleted by ultracentrifugation (100k rpm for 1 h at 4 °C in a TLA100.3 rotor). The ribosome pellet was resuspended in High Salt Buffer (50 mM KHEPES pH 7.5, 1 M KOAc, 15 mM Mg(OAc)$_2$, 0.1% Triton, 2 mM DTT) to ~1 μM and incubated with 1 μM tetrazine-conjugated BDP (Jena Bioscience) at 25 °C for 20 min. The labeled RNC was purified as described for unpurified RNC and stored at –80 °C.

**Preparation of HSW**
HSW(RRL). Raw RRL was ultracentrifuged at 100k rpm in a TLA100.3 rotor for 1 h at 4 °C to pellet the ribosome. The ribosome pellet was resuspended in 0.1× original volume of Low Salt Buffer (50 mM KHEPES pH 7.5, 50 mM KOAc, 1 mM Mg(OAc)$_2$, 2 mM DTT) and centrifuged at 14k rpm in an Eppendorf 5425 rotor for 10 min at 4 °C to remove large aggregates. The supernatant was layered on a 0.5 M sucrose cushion in Low Salt buffer at 1:2 volume ratio and ultracentrifuged at 100k rpm in a TLA100.3 rotor for 1 h at 4 °C. The ribosome pellet was resuspended at ~2 μM in Low Salt Buffer. The salt concentration of the ribosome suspension were adjusted to 750 mM KOAc and 15 mM Mg(OAc)$_2$, and the solution was incubated at 4 °C for 1 h followed by ultracentrifugation at 100k rpm in a TLA100.3 rotor for 0.5 h at 4 °C. The supernatant was dialyzed into Assay buffer and stored as HSW(RRL) at –80 °C. The concentration of HSW was defined as the amount equivalent to the ribosome from which the HSW was prepared.

HSW(WG). Wheat germ lysate was prepared following the method described by Erickson et al.[72]. Commercial raw wheat germ (Fearn) was ground in liquid nitrogen using a mortar and a pestle to fine powder. Once the liquid nitrogen has evaporated, the wheat germ powder was transferred to a second mortar and ground again in ice cold homogenization buffer (50 mM KHEPES pH 7.5, 100 mM KOAc, 1 mM Mg(OAc)$_2$, 2 mM CaCl$_2$, 2 mM DTT). The homogenate was centrifuged at

20,000 rpm in a JA20 rotor for 10 min at 4 °C. Clarified supernatant was stored at –80 °C as wheat germ lysate. HSW(WG) was prepared the same procedures as for HSW(RRL), except that the wheat germ lysate was used as the starting material.

**Biochemical measurements**. To remove aggregates, RNC and SRP were centrifuged at 14k rpm in an Eppendorf 5425 rotor for 30 min at 4 °C, and SR, NAC and HSW were centrifuged at 100k rpm in a TLA100 rotor for 30 min at 4 °C before all assays. All measurements were carried out at 25 °C in Assay buffer supplemented with 1 mg/mL BSA to prevent non-specific adsorption to surfaces. All reported standard deviations (SDs) are calculated from separated measurements on different samples.

SRP-SR association kinetics. RNC (400 nM) was pre-incubated with SRP$^{Cy3B}$ (20 nM) and, where indicated, with NAC (2× final concentration) or HSW (equivalent to the amount prepared from 400 nM ribosome) in Assay Buffer with 1 mg/mL BSA and 2 mM GTP. In parallel, Atto647n-labeled SR(R458A), a GTPase-deficient mutant of SR[10], was prepared at 2× of the final concentration in Assay Buffer with 1 mg/mL BSA and 2 mM GTP. The two solutions were mixed in equal volume on a stopped-flow apparatus (Kintek) to initiate the reaction. SRP-SR association was monitored by recording the fluorescence intensity of Cy3B excited at 535 nm using a 580/20 nm optical filter and a photo-multiplier tube. The time traces were fitted to exponential decay functions to extract the observed association rate constant, $k_{obs}$. Plots of $k_{obs}$ as a function of SR concentration were fit to Eq. (1),

$$k_{obs} = k_{on}[SR] + k_{off} \tag{1}$$

in which $k_{on}$ is the association rate constant between SRP and SR, and $k_{off}$ is the dissociation rate constant of the SRP-SR complex.

Steady-state fluorescence measurements. To detect FRET between BDP-labeled RNC (RNC$^{BDP}$) and TMR-labeled SRP (SRP$^{TMR}$), fluorescence emission spectra were recorded for RNC$^{BDP}$ (1 nM), RNC$^{BDP}$ mixed with 5 nM SRP$^{TMR}$, and 50 nM unlabeled SRP added to the preformed RNC$^{BDP}$•SRP$^{TMR}$ complex. To detect FRET between RNC$^{BDP}$ and Cy3B-labeled NAC (NAC$^{Cy3B}$), fluorescence emission spectra were recorded for RNC$^{BDP}$ (1 nM), RNC$^{BDP}$ mixed with 100 nM NAC$^{Cy3B}$, and 500 nM unlabeled NAC added to the preformed RNC$^{BDP}$•NAC$^{Cy3B}$ complex. Spectra were recorded on a Fluorolog 3-22 spectrometer (Jobin Yvon) using an excitation wavelength of 485 nm.

For equilibrium titrations to measure the binding affinity of NAC or SRP for the RNC, ~0.01 volumes of SRP$^{TMR}$ or NAC$^{Cy3B}$ stock solutions were mixed with 300 μL RNC$^{BDP}$ (1 nM in SRP titrations and 5 nM in NAC titrations) in Assay Buffer with 1 mg/mL BSA to reach the indicated final titrant concentrations. Where indicated, unlabeled NAC or SRP was premixed with RNC at the specified concentrations. The fluorescence intensity of BDP was measured using an excitation wavelength of 485 nm and an emission wavelength of 517 nm. Raw fluorescence intensity readings were corrected for dilutions during the titration, and FRET efficiency was calculated using Eq. (2),

$$E = 1 - \frac{F_{DA}}{F_D} \tag{2}$$

in which $E$ is FRET efficiency, $F_{DA}$ and $F_D$ are the fluorescence intensities of the donor dye with or without acceptor present, respectively.

For competition titration with 80S, 2.6 nM RNC RNC$^{BDP}$, and 4 nM SRP$^{TMR}$ were mixed together in Assay Buffer with 1 mg/mL BSA to reach equilibrium. 80S was titrated into the solution and the fluorescence intensity of BDP was measured using an excitation wavelength of 485 nm and an emission wavelength of 517 nm.

**Single-molecule TIRF with alternating laser excitation**. RNC (100 nM) translated on 3′-biotinylated mRNA was incubated with doubly labeled SRP (10 nM) for 5 min at 25 °C and diluted 20-fold in Image Buffer (Assay Buffer supplemented with 1 mg/mL BSA, 200 μM non-hydrolyzable GTP analog Guanosine-5′-[(β,γ)-imido]triphosphate (GppNHp), 4 mM Trolox, and GODCAT oxygen scavenge system[73]) with or without 300 nM NAC. The solution was loaded onto quartz slides passivated with PEGylation[74]. Movies were recorded using MicroManager on a home-built system as described before[75] with alternating excitation using the donor (532 nm) and acceptor (635 nm) lasers at a frame rate of 10 s$^{-1}$. The single-molecule movies were analyzed with iSMS[76].

For single-molecule colocalization between NAC and SRP, RNC (1.5 nM) translated on 3′-biotinylated mRNA in Image buffer was loaded onto quartz slides passivated with PEGylation and incubated for 10 min at 25 °C. The RNC-coated slide chamber was washed with Image buffer to remove unbound RNC. NAC$^{Cy3B}$ (2 nM) and SRP$^{Atto647n}$ (1 nM) were mixed in Image buffer and loaded on to the slide. For negative controls, no RNC was immobilized on the slide surface (no RNC) or RNC(ss) was immobilized with NAC$^{Cy3B}$ replaced by NACmt$^{Cy3B}$ (RNC (ss) w/ NACmt) The data acquisition and analysis were the same as single-molecule FRET experiment.

**Miscellaneous biochemistry**
Cotranslational protein translocation assay. The cotranslational targeting and translocation of preproteins were measured as described previously[10,33]. mRNA encoding pPL or pPL variants were translated in wheat germ lysate (Promega) containing [$^{35}$S]-methionine for 2 min, followed by addition of purified SRP

(indicated concentrations in SRP titration and 50 nM in SR titrations), SR (74 nM in SRP titration and indicated concentrations for SR titration), and trypsine-digested, salt-washed rough ER microsome (0.5 eq/μL)[77]. The reactions were incubated at 25 °C for 40 min and quenched with 2× SDS sample buffer. Translation products were separated on SDS-PAGE, and translocation efficiency was calculated using the following equation:

$$\text{Translocation efficiency} = \frac{\frac{8}{7} \times \text{prolactin}}{\frac{8}{7} \times \text{prolactin} + \text{preprolactin}} \quad (3)$$

in which the factor 8/7 accounts for the numbers of methionines in preprolactin and prolactin.

**Optimization of amber suppression.** In vitro translation reactions were carried out in RRL with indicated concentration of PyltRNA, PylRS, and TCOK. Methionine in the reaction was replaced with [$^{35}$S]-methionine. After incubation at 32 °C for 30 min, the reaction was quenched with 2× SDS sample buffer and analyzed by SDS-PAGE and autoradiography.

**Western blot.** RNC, HSW(RRL) and recombinantly purified NAC were loaded onto SDS-PAGE at indicated concentrations, Western blotted with anti-NACβ antibody (Abcam, EPR16495) using dilution of 1:1000, and detected using IRDye 800CW goat anti–rabbit IgG (925-32211; LI-COR Biosciences) using dilution of 1:10,000. Western blot signals were quantified using the Odyssey CLx imaging system (LI-COR Biosciences).

Gel image processing was done using ImageJ.

**Equilibrium titration fitting.** For SRP titrations, observed FRET efficiencies were plotted as a function of SRP concentration and fit to Eq. (4),

$$E = E_{\text{SRP}} \times \frac{[\text{SRP}]}{K_{\text{d,SRP}} + [\text{SRP}]} \quad (4)$$

in which $E_{\text{SRP}}$ is the FRET efficiency when RNC$^{\text{BDP}}$ is bound by SRP$^{\text{TMR}}$, and $K_{\text{d,SRP}}$ is the dissociation constant of the RNC-SRP complex. For NAC titrations, values of $E$ were plotted against NAC concentration and fit to Eq. (5),

$$E = E_{\text{NAC}}$$
$$\times \frac{K_{\text{d,NAC}} + [\text{RNC}]_0 + [\text{NAC}] - \sqrt{(K_{\text{d,NAC}} + [\text{RNC}]_0 + [\text{NAC}])^2 - 4[\text{RNC}]_0[\text{NAC}]}}{2[\text{RNC}]_0} \quad (5)$$

in which $E_{\text{NAC}}$ is the FRET efficiency when RNC$^{\text{BDP}}$ is bound by NAC$^{\text{Cy3B}}$, [RNC]$_0$ is the added concentration of RNC$^{\text{BDP}}$ (5 nM), and $K_{\text{d,NAC}}$ is the dissociation constant of the RNC-NAC complex.

For competition experiments to measure the binding of unlabeled NAC to RNC, RNC$^{\text{BDP}}$ (1 nM), and NAC$^{\text{Cy3B}}$ (20 nM) were pre-incubated in Assay Buffer with 1 mg/mL BSA. Aliquots of 0.01× volume of unlabeled NAC stock solution was added to reach the indicated final NAC concentrations. The fluorescence intensity of RNC$^{\text{BDP}}$ was corrected for dilution, and FRET efficiency was calculated using Eq. (2). The data were plotted against the concentration of unlabeled NAC and fit to Eq. (6),

$$E = E_{\text{NAC}} \times \frac{1}{1 + \frac{K_{\text{d,L}}}{[\text{NAC}^L]_0} + \frac{K_{\text{d,L}}}{[\text{NAC}^L]_0} \times \frac{[\text{NAC}^U]}{K_{\text{d,U}}}} \quad (6)$$

in which $E_{\text{NAC}}$ is defined above, $K_{\text{d,L}}$ and $K_{\text{d,U}}$ are the $K_{\text{d}}$'s of RNC$^{\text{BDP}}$ for NAC$^{\text{Cy3B}}$ and unlabeled NAC, respectively, and [NAC$^L$]$_0$ and [NAC$^U$] are the concentrations of NAC$^{\text{Cy3B}}$ (20 nM) and unlabeled NAC, respectively.

For competition binding between RNC(ss) and 80S to SRP with or without NAC, the titration results were fitted to a competition equation:

$$F = 1 - \frac{1}{1 + \frac{[80\text{S}]}{K_i}} \quad (7)$$

in which $F$ is the normalized fluorescence change, [80S] is the concentration of 80S ribosome and $K_i$ is the competition coefficient.

For global fitting with the anti-cooperative model, the RNC-SRP titration data across all concentrations of NAC were simultaneously fit to Eq. (8),

$$E = \frac{E_{\text{SRP}} + E_{\text{SRP,NAC}} \times \frac{[\text{NAC}]}{\alpha \times K_{\text{d,NAC}}}}{1 + \frac{K_{\text{d,SRP}}}{[\text{SRP}]} + \frac{[\text{NAC}]}{[\text{SRP}]} \times \frac{K_{\text{d,SRP}}}{K_{\text{d,NAC}}} + \frac{[\text{NAC}]}{\alpha \times K_{\text{d,NAC}}}} \quad (8)$$

in which $E_{\text{SRP,NAC}}$ is the FRET efficiency when both SRP$^{\text{TMR}}$ and NAC are bound to RNC$^{\text{BDP}}$. $E_{\text{SRP}}$, $K_{\text{d,SRP}}$, and $K_{\text{d,NAC}}$ are defined above.

For global fitting with the competitive model, the same data were simultaneously fit to Eq. (9),

$$E = \frac{E_{\text{SRP}}}{1 + \frac{K_{\text{d,SRP}}}{[\text{SRP}]} + \frac{[\text{NAC}]}{[\text{SRP}]} \times \frac{K_{\text{d,SRP}}}{K_{\text{d,NAC}}}} \quad (9)$$

To compare the experimental $K_{\text{d,SRP}}$ and $K_{\text{d,NAC}}$ values to predictions from the different models, the predicted NAC concentration dependence of the apparent $K_{\text{d,SRP}}$ values ($K_{\text{d}}$) from the anti-cooperative and competitive models were simulated using Eq. (10) and Eq. (11), respectively.

$$K_{\text{d}} = K_{\text{d,SRP}} \times \frac{K_{\text{d,NAC}} + [\text{NAC}]}{K_{\text{d,NAC}} + \frac{[\text{NAC}]}{\alpha}} \quad (10)$$

$$K_{\text{d}} = K_{\text{d,SRP}} \times \left(1 + \frac{[\text{NAC}]}{K_{\text{d,NAC}}}\right) \quad (11)$$

Reciprocally, the predicted SRP concentration dependence of the apparent $K_{\text{d,NAC}}$ values ($K_{\text{d}}$) from the anti-cooperative and competitive models were simulated using Eq. (12) and Eq. (13), respectively.

$$K_{\text{d}} = K_{\text{d,NAC}} \times \frac{K_{\text{d,SRP}} + [\text{SRP}]}{K_{\text{d,SRP}} + \frac{[\text{SRP}]}{\alpha}} \quad (12)$$

$$K_{\text{d}} = K_{\text{d,NAC}} \times \left(1 + \frac{[\text{SRP}]}{K_{\text{d,SRP}}}\right) \quad (13)$$

**Analysis of single-molecule fluorescence data.** FRET efficiency ($E$) and stoichiometry ($S$) were calculated from raw fluorescence time traces using Eq. (14) and Eq. (15), respectively,

$$E = \frac{F_{\text{DD}}}{F_{\text{DD}} + F_{\text{AD}}} \quad (14)$$

$$S = \frac{F_{\text{DD}} + F_{\text{AD}}}{F_{\text{DD}} + F_{\text{AD}} + F_{\text{AA}}} \quad (15)$$

where $F_{\text{DD}}$ and $F_{\text{AD}}$ are the fluorescence emission intensities of the donor and acceptor dyes with donor excitation, and $F_{\text{AA}}$ is the emission intensity of the acceptor dye with acceptor excitation. The smFRET histograms were fit to the sum of three-Gaussian functions using Eq. (16),

$$\text{PDF} = \sum_{i=1}^{3} A_i \times \frac{1}{\sigma_i \sqrt{2\pi}} e^{-\frac{1}{2}\left(\frac{E - \mu_i}{\sigma_i}\right)^2} \quad (16)$$

in which PDF is the population density function, $A_i$ and $\sigma_i$ are the weight and standard deviation of the $i$-th Gaussian, respectively. $\mu_i$ is the center of the $i$-th Gaussian and is indicated in the figures. $A_i$ is further plotted in a cumulative bar graph to show the proportion of each FRET state.

For analysis of dwell time for colocalization experiment, fluorescence traces were fitted to Hidden Markov Model (HMM) with built-in function of iSMS. Colocalization was defined based on signal above background in $A_{\text{em}}$–$D_{\text{ex}}$ for RNC (ss) and $D_{\text{em}}$–$D_{\text{ex}}$ for RNC(ssmt). The cumulative probability distribution of dwell time in colocalization state was fitted to a two-exponential function

$$f(t) = A_{\text{fast}} \times e^{-k_{\text{fast}}t} + A_{\text{slow}} \times e^{-k_{\text{slow}}t} \quad (17)$$

to extract the kinetic parameters. $k_{\text{fast}}$ and $k_{\text{slow}}$ are the rate constants for the fast and slow phases, respectively. $A_{\text{fast}}$ and $A_{\text{slow}}$ are the weights for the fast and slow phases, respectively.

**Model of SRP-dependent co-translational protein targeting.** The experimental $k_{\text{on,SR}}$ values for SRP on RNC(ss) were plotted as a function of NC length and fit to Eq. (18),

$$k_{\text{on}} = k_{\text{on,min}} \times \left(\frac{1}{1 + Ae^{\frac{\text{ss}}{L}}}\right) + k_{\text{on,max}} \times \left(1 - \frac{1}{1 + Ae^{\frac{\text{ss}}{L}}}\right) \quad (18)$$

in which ss is the exposed length of signal sequence (0–14 aa), $k_{\text{on,min}}$ and $k_{\text{on,max}}$ are the minimal and maximal $k_{\text{on}}$, respectively, $L$ is the characteristic length of the hydrophobic sequence needed to activate SRP-SR association, and $A$ is the scaling factor that quantifies how sensitive the system is to an exposed hydrophobic sequence. The parameters obtained from this fit are: $k_{\text{on,min}} = 4.0 \times 10^5$ M$^{-1}$s$^{-1}$, $k_{\text{on,max}} = 9.4 \times 10^5$ M$^{-1}$s$^{-1}$, $A = 2.8 \times 10^{-3}$ and $L = 1.1$ aa without NAC; $k_{\text{on,min}} = 2.2 \times 10^4$ M$^{-1}$s$^{-1}$, $k_{\text{on,max}} = 1.3 \times 10^6$ M$^{-1}$s$^{-1}$, $A = 8.2 \times 10^{-4}$ and $L = 2.2$ aa with NAC present. These values were plugged into Eq. (18) to calculate the $k_{\text{on,SR}}$ values at arbitrary NC lengths for SRP bound to RNC(ss). The values of $k_{\text{on,SR}}$ for SRPs on RNC(ssmt) were independent of NC length and set to $1.3 \times 10^4$ and $5.7 \times 10^5$ M$^{-1}$s$^{-1}$ with and without NAC, respectively.

In all modeling, SRP and SR concentrations were both set to 500 nM, which are the estimated in vivo concentrations from quantitative mass spectrometry data[16]. Translation elongation rate ($k_{\text{elongation}}$) was estimated to be 6 aa s$^{-1}$ according to a recent ribosome profiling data[43]. $K_{\text{d,SRP}}$ for the RNC•SRP complex were estimated to be 4 nM based on the measurements in this work. The $k_{\text{d,SRP}}$ and $k_{\text{off,SRP}}$ values were set to be $10^7$ M$^{-1}$s$^{-1}$ and 0.04 s$^{-1}$ to satisfy the estimated $K_{\text{d,SRP}}$ values. As shown in the sensitivity tests in Fig. 7e–g, the values of $K_{\text{d,SRP}}$ and $k_{\text{on,SRP}}$ values can vary by up to two orders of magnitude without affecting protein targeting efficiency. The rate constant for commitment of the RNC-SRP-SR complex to translocation ($k_{\text{target}}$) was estimated to be 0.2 s$^{-1}$, given that the fastest rate of GTP hydrolysis in the SRP-SR complex is ~0.2 s$^{-1}$ and that RNC must engage Sec61p for translocation before the SRP•SR complex is disassembled through GTP

hydrolysis[10]. Values of $k_{off,SR}$ were calculated from $K_d \times k_{on}$ assuming a $K_{d,SR}$ value of 100 nM, based on previous measurements of SRP-SR binding affinities on the ribosome, RNC(ss) and RNC(ssmt)[10].

Co-translational protein targeting was determined using the following differential equations that describe the model in Fig. 7b:

$$\frac{d}{dt}[RNC_i] = k_{elongation}([RNC_{i-1}] - [RNC_i]) - k_{on,SRP}[SRP]_0[RNC_i]$$
$$+ k_{off,SRP}[RNC_i \cdot SRP] \tag{19}$$

$$\frac{d}{dt}[RNC_i \cdot SRP] = k_{elongation}([RNC_{i-1} \cdot SRP] - [RNC_i \cdot SRP]) + k_{on,SRP}[SRP]_0[RNC_i]$$
$$- (k_{off,SRP} + k^i_{on,SR}[SR]_0)[RNC_i \cdot SRP] + k^i_{off,SR}[RNC_i \cdot SRP \cdot SR] \tag{20}$$

$$\frac{d}{dt}[RNC_i \cdot SRP \cdot SR] = k_{elongation}([RNC_{i-1} \cdot SRP \cdot SR] - [RNC_i \cdot SRP \cdot SR])$$
$$+ k^i_{on,SR}[SR]_0[RNC_i \cdot SRP] - (k^i_{off,SR} + k_{target})[RNC_i \cdot SRP \cdot SR] \tag{21}$$

$$\frac{d}{dt}\left[RNC_{i,targeted}\right] = k_{elongation}\left(\left[RNC_{i-1,targeted}\right] - \left[RNC_{i,targeted}\right]\right)$$
$$+ k_{target}[RNC_i \cdot SRP \cdot SR] \tag{22}$$

$RNC_i$ is the RNC carrying a NC of length $i$, the values for $[SRP]_0$, $[SR]_0$, $k_{elongation}$, $k_{target}$, $k_{on,SRP}$, and $k_{off,SRP}$ are listed in Fig. 7d and explained above.

The differential equations were solved for the population of each state of RNC at different NC lengths using the algorithm developed by Sharma et al.[44]. Specifically, the chemical kinetics master equations above can be described by a form of linear equation:

$$\frac{d}{dt}P_n(i) = k_{elongation}(P_n(i-1) - P_n(i)) + \sum_{m=1,m\neq n}^{N} k_{mn}(i) \times P_m(i)$$
$$- \sum_{m=1,m\neq n}^{N} k_{nm}(i) \times P_n(i) \tag{23}$$

where $n$ is the indices for the species of RNC (RNC, SRP-RNC, SRP-SR-RNC, and targeted RNC), $P_n(i)$ is the population of species $n$ at a nascent chain length $i$ and $k_{mn}$ is the first-order rate constant for the conversion of species $m$ to species $n$. These linear equations can be described in the matrix form:

$$\frac{d}{dt}\mathbf{P}(i) = \mathbf{A} \times \mathbf{P}(i-1) - \mathbf{A} \times \mathbf{P}(i) + \mathbf{T} \times \mathbf{P}(i) \tag{24}$$

where

$$\mathbf{P}(i) = \begin{bmatrix} RNC_i \\ RNC_i \cdot SRP \\ RNC_i \cdot SRP \cdot SR \\ RNC_{i,targeted} \end{bmatrix}, \mathbf{A} = \begin{bmatrix} k_{elongation} & 0 & 0 & 0 \\ 0 & k_{elongation} & 0 & 0 \\ 0 & 0 & k_{elongation} & 0 \\ 0 & 0 & 0 & k_{elongation} \end{bmatrix} \tag{25}$$

$$\mathbf{T} = \begin{bmatrix} -k_{on,SRP}[SRP]_0 & k_{off,SRP} & 0 & 0 \\ k_{on,SRP}[SRP]_0 & -k_{off,SRP} - k^i_{on,SR}[SR]_0 & k^i_{off,SR} & 0 \\ 0 & k^i_{on,SR}[SR]_0 & -k^i_{off,SR} - k_{target} & 0 \\ 0 & 0 & k_{target} & 0 \end{bmatrix} \tag{26}$$

To calculate the steady-state population, one solves the linear equation:

$$\frac{d}{dt}\mathbf{P}(i) = \mathbf{A}\mathbf{P}(i-1) - \mathbf{A}\mathbf{P}(i) + \mathbf{T}\mathbf{P}(i) = 0 \tag{27}$$

and gets:

$$\mathbf{P}(i) = [\mathbf{A} - \mathbf{T}(i)]^{-1} \times \mathbf{A} \times \mathbf{P}(i-1) \tag{28}$$

where $\mathbf{P}(i)$ represents the population of each state at NC length of $i$. The initial condition $\mathbf{P}(1)$ was set to:

$$\mathbf{P}(1) = \begin{bmatrix} 1 \\ 0 \\ 0 \\ 0 \end{bmatrix} \tag{29}$$

which means that there is only free RNC at a NC length of 1. The populations of species at subsequent nascent chain lengths were calculated by propagating $\mathbf{P}(1)$ using Eq. (28).

**Reporting summary**. Further information on research design is available in the Nature Research Reporting Summary linked to this article.

## Data availability
Data supporting the findings of this manuscript are available from the corresponding author upon reasonable request. Structural data associated with Fig. 1a are available in the Protein Data Bank under accession code 4UG0 (ribosome)[78]; and in Electron Microscopy Database under accession codes: EMD-3037 (SRP)[6]; EMD-4938 (NAC)[23]; EMD-6105 (RAC)[79]; and EMD-0202 (NatA/E)[80]. Source data are provided with this paper.

## Code availability
The scripts for modeling of cotranslational protein targeting are available at GitHub: https://github.com/emc2emc2/2020_cotranslational_targeting.

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

## Acknowledgements

We thank J. W. Chin for sharing PyltRNA/RS plasmid, D. Baltimore for sharing EMCV IRES sequence, A. Hoelz for sharing PreScission protease, R. M. Voorhees for advice on RRL in vitro translation and RNC purification, R. L. Gonzalez Jr for advice on the single-molecule TIRF setup, and members of the Shan lab for discussions and advice on this work. This work was supported by National Institutes of Health grant R01 GM078024, R35 GM136321, NSF grant MCB-1929452, and the Gordon and Betty Moore Foundation through grant GBMF2939 to S.-o.S.

## Author contributions

H.-H.H. and S.-o.S. conceived and designed the study. H.-H.H., J.H.L., and S.C. expressed and purified proteins and RNAs. H.-H.H. performed experiments, analyzed data, and performed simulations. H.-H.H. and S.-o.S. prepared the figures and wrote and edited the manuscript. S.-o.S. supervised the project and secured funding.

## Competing interests

The authors declare no competing interests.
