## [Peer Review File · Nature Communications]

REVIEWER COMMENTS

Reviewer #1 (Remarks to the Author):

This study by the group of Shan addresses the question of how cells differentiate between proteins that must be exported from the cell versus those that must remain inside the cell. Signaling sequences, SRP, and SRP receptor are all necessary components for targeting these proteins to the ER, but they are not sufficient. With these components, even nascent proteins that lack signal sequences will tend to be targeted to the ER above the level observed in the cell. In this study, Shan and co-workers demonstrate that the NAC complex is also needed to further differentiate between these two classes of proteins. And that the most likely mechanism by which this occurs is through the co-binding of both NAC and SRP to ribosomes, with NAC tending to switch SRP into a non-functional state when nascent proteins are present that lack signal sequences. And that this reduces the affinity of the ribosome complex for the SRP receptor, thereby reducing improper protein targeting to the ER. This is an excellent study, an important new discovery, and advances our understanding of how the ensemble of protein biogenesis factors act together to lead to proper protein maturation.

The only major concern I have is with the assumption in the mathematical model that K_d for SRP is constant at all nascent chain lengths. Experiments from a number of groups show that K_d is length dependent – it starts high at short nascent chain length, goes low, then increases again at the longest lengths. How will incorporating this behavior affect the results? This should be addressed. (Note well, varying K_d as they did in panel 6f is not sufficient, as I understand that here they still kept the assumption of K_d independent of nascent chain length.)

I have the following minor suggestions:

1. The use of the 'simulation' is not appropriate when describing their solutions to the mathematical model. Rather, they 'numerically solved' the model. When you have an analytical function that you solve without any stochastic technique then it is not a simulation, the only uncertainty comes from the numerical method itself. For example, with the authors' technique, for the same arguments you always get the same solution. But for Monte Carlo, or Molecular dynamics you get different results. Examples of how the text can be altered, line 396: "to be simulated" -> "to be determined"; 399: "The simulation showed" -> "The model exhibits", etc.

2. A perspective on the interplay of timescales during protein biogenesis was written last year that should be cited PMID: 29414517 as it provides a broader context that the current manuscript fits into.

Reviewer #2 (Remarks to the Author):

Review of Hsieh et al. "A ribosome-associated chaperone enables substrate triage in a co-translational protein targeting complex"

The manuscript of Hsieh et al. is a deep, rigorous investigation of a key question in protein biogenesis—how are proteins co-translationally targeted to the ER by the SRP-receptor system with appropriate specificity? The authors tackle a key unanswered question in eukaryotic protein targeting—what determines the specificity for polypeptides for a signal-sequence versus those that do not; the SRP-receptor-signal peptide interactions on their own do not explain the observed in vivo targeting specificity. A possible co-factor in establishing fidelity of targeting is the abundant ribosome-binding chaperone NAC, which was thought to compete with SRP to increase specificity for appropriate signal sequence targeting. Here the authors upend this notion, and show conclusively that NAC binds (slightly anticooperatively) to ribosome-nascent chain-SRP complexes, and modulate the association rates of the complexes for the SRP receptor (RS), especially disfavoring the binding of ribosomes with short or non-signal peptides. The authors carry out careful kinetic targeting experiments using a

combination of biochemical approaches (targeting kinetics in translation extracts with liver microsomes, and in particular careful bulk and single-molecule fluorescence assays to monitor SRP-ribosomal nascent chain (RNC) interaction, NAC-SRP-RNC interaction, and RS-SRP interactions, using a variety of nascent chain sequences and lengths. The results are rate and equilibrium constants for the key interactions and thermodynamic coupling parameters for NAC and SRP (they are negatively coupled, meaning binding of one makes binding of the partner less stable). They then perform challenging single-molecule FRET experiments to monitor the conformation of the SRP on an RNC, using doubly-labeled SRP. These experiments showed that SRP bound to a correct signal sequence RNC has largely a single conformation with high FRET, and addition of NAC shifts this conformational distribution, whereas with an incorrect signal sequence, the FRET distributions are broad (3 gaussian fit) and NAC shifts these distributions almost entirely into a single low FRET state. The authors interpret these results to show how NAC prepares an SRP-RNC complex for proper targeting, preventing the targeting to RS for incorrect or too short signal sequences. They support this model through a kinetic scheme and simulation, using their experimental data. The model is consistent with *in vivo* investigations, and should stimulate a number of additional *in vivo* and biophysical experiments. This is challenging biophysics of the first order, and as such the manuscript should be published in Nature Communications, but first I would like to see the authors address several points.

1. It would be nice to confirm co-occupancy of NAC and SRP. Have they performed direct FRET experiments to show that both factors reside on the ribosome? Their indirect data are supportive, but direct demonstration would be better.
2. For the single-molecule FRET data, are the timescales of the conformational changes for the mutant signal sequence, or NAC-induced changes? Are these rapid, or can state transitions be observed in the data. The authors should at least discuss this. Also, what is the molecular interpretation of these different states? Are their hints from structural data? I'm not sure that panel 5e is needed; it was confusing on first glance
3. What is the dissociation rate of NAC from the ribosome? Was it determined independently, or we just have equilibrium constants?
4. The final model figure (7) should be improved. I had a difficult time to see the differences in the nascent peptides from real ss to weak, to too short. Some suggestions--Perhaps thicken the lines for the peptides, and do not use the gray shaded background. Also, it would be nice to have the reaction arrows for all three states show explicitly or the molecules grouped. It looks like only the central strong sequence is undergoing the reaction in the no NAC case.

Reviewer #3 (Remarks to the Author):

In their report Hsieh et al have studied the influence of the nascent-chain associated complex (NAC) on the fidelity of the co-translational protein targeting pathway by the Signal recognition particle (SRP). The authors conclude that the mammalian SRP and its receptor SR are unable to provide the required specificity for protein targeting to the ER membrane. Rather, they propose that by binding to SRP, NAC regulates the SRP-SR interaction and thus prevents the targeting of ribosome-nascent chain complexes (RNCs) with incorrect or short signal sequences. Thus, NAC would serve a crucial role in maintaining targeting fidelity.

The conclusion that NAC is a modulator of SRP specificity is actually not really new and has been already proposed earlier based on biochemical assays (Zhang et al., 2012) and on ribosome profiling data (del Alamo et al., 2011). However, considering the fact that the cellular NAC concentration equals the ribosome concentration, while SRP is largely sub-stoichiometric (approx. 1/100), it appears rather unlikely that the function of NAC is restricted to modulating the SRP function on the ribosome.

The initially suggested competitive binding model to which the authors refer (Wiedmann et al., 1994),

which had suggested that NAC would prevent binding of incorrect cargos to the Sec61 complex at the ER membrane, was also already questioned by several studies (e.g. Raden & Gilmore, 1998; Neuhof et al., 1998). What is indeed correct and what has recently been further validated is that SRP and NAC engage almost identical binding sites on the ribosome and that they are both able to protrude into the ribosomal tunnel (Jomaa et al. 2016; Gamerdinger et al., 2019).

Nevertheless, the authors data further support the hypothesis that NAC modulates the SRP-dependent targeting and they conclude that NAC 'remodels the conformational landscape of SRP on the ribosome'. Although this is possibly true, the statement itself is rather vague and does not reveal any molecular mechanism. Thus, in summary, the manuscript contains interesting and rather complex data, but it does not really answer the question of how NAC improves targeting fidelity by the SRP pathway.

In addition, I have several issues the authors need to address for improving their study.

1. The authors use pre-prolactin (pPL) with different engineered signal sequences as substrates for their assays. Fig. 1c shows in vitro translocation of these substrates into microsomes and different to what the authors state, pPL containing the synthetic signal sequence (ss) is less efficiently translocated than pPL containing the wild type signal sequence. Yet, pPL containing the synthetic signal sequence is used in all experiments as reference. The rationale for this is not clear to me and the use of an already 'suboptimal' substrate as positive control is likely causing some bias in the interpretation. Therefore, the use of wild type pPL as positive control would be a much better choice.
2. lane 137/138: the authors state that SRP rejects non-optimal substrates. This statement is not correct. All the assay shows is that signal sequence cleavage of the ss, ssmt and ssmt2 constructs is reduced. Whether this is due to rejection by SRP, by the Sec61 complex or by signal peptidase cannot be deduced from these assays and would require more detailed analyses.
3. The assays make use of a soluble SRP receptor and as the authors state, the absence of the transmembrane domain of SR β has only little effect on cell growth or protein translocation in yeast cells (Ogg et al., 1998). However, whether this also applies to the mammalian system is not known. The available data rather indicate that data from yeast cannot simply be transferred to the mammalian system without further proof. This is exemplified by the fact that neither SRP nor NAC are essential in yeast (or cells can at least cope with their absence), while both proteins are strictly essential in mammalian cells.
It is also important to emphasize that later work had shown that even in yeast the absence of the transmembrane domain impairs gating of the Sec61 complex (Jiang et al., 2007) and that contact between SR β and Sec61 is important for SRP release from the signal sequence (Fulga et al., 2001). Thus, the use of a soluble SR in an assay that is supposed to monitor membrane targeting is likely causing some additional bias.
4. Lane 154: as mentioned above, rejection is not shown by the assays depicted in Fig. 1 and the ssmt2 construct has a much stronger effect on SR-SRP complex formation (Fig. 2). This demonstrates that the SRP system has an intrinsic proof-reading activity even in the absence of any additional factor, like NAC. This is also obvious from the data shown in 2g.
5. Lane 197-199: This statement is misleading because the experimental system the authors are using cannot differentiate between early and late steps of the targeting reaction. SRP-SR interaction in vivo is a later step of targeting but the authors use the SRP-SR interaction as read-out for substrate recognition by SRP, which occurs before. Therefore, if the authors are indeed interested in understanding the fidelity of SRP to recognize its substrates, then monitoring the SRP-SR interaction is not the best read-out and monitoring SRP-RNC interaction directly as shown in Fig. 3 is a much better way.
6. Fig. 3 shows that SRP binds with nanomolar affinity to both RNC(ss) and RNC(ssmt). Here it would be important to see also the affinity for RNC(ssmt2), because the latter is indeed an incorrect substrate and should show similar nanomolar affinity if SRP indeed requires NAC for correct substrate recognition/rejection of incorrect substrates.
7. Why does NAC reduce the SRP-RNC(ss) interaction, if the ss signal sequence is considered to reflect

a correct substrate (Fig. 3). This supports my earlier concern about using ss and these experiments should be performed with constructs harboring the native signal sequence.

8. Lane 258: I am not sure how to understand the statement 'SRP and NAC modestly weaken the binding of each other to the RNC'. 'Modest weakening' does not seem to be compatible with the main conclusion of the manuscript that NAC enables substrate triage.

I also think that a competitive binding model cannot be excluded. Considering that both SRP and NAC protrude into the ribosomal tunnel in non-translating ribosomes or at very early stages of translation (Jomaa et al., 2016, Gemerdinger et al., 2019, Denks et al., 2017; Mercier et al., 2019), competition for ribosome binding can still occur at early stages but could then be followed by cooperative interaction when significant portions of the nascent chain have emerged from the ribosomal tunnel.

9. Fig. 4 shows that NAC cannot differentiate between RNC(ss) and RNC(ssmt). Again, it would be important to show the ssmt2 RNCS for analyzing whether NAC has any ability to discriminate between SRP and non-SRP signal sequences.

10. Fig. 4a is also not properly designed: while it shows the decrease in donor fluorescence it does not show the corresponding increase in acceptor fluorescence, therefore the data cannot be properly interpreted.

11. The smFRET approach is nice, but what is the biological significance behind the low, medium and fast FRET populations? Also why is the medium-FRET population less capable of interacting with SR? As these data stand right now, they don't provide much insight into the process, but are rather descriptive on a very fancy level.

12. Lane 354: different to what the authors state, signalless RNCs were not tested only RNCS with incorrect signal sequences.

13. For addressing whether NAC influences the pre-emptive targeting of RNCs harboring short nascent chains, it would be much better to monitor the direct SRP-RNC interaction as in Fig. 3 (putting the label on the ribosome) instead of testing the rather indirect SRP-SR interaction. This would directly determine whether SRP binding to very short RNCs or even non-translating ribosomes is influenced by NAC.

Dear editor,

Thank you for your time and work handling this manuscript. We also thank the reviewers for their constructive comments, which helped making this manuscript stronger. Below are our point-by-point responses (blue) to the reviewers' comments (in black). We hope that the revised manuscript is now suitable for publication in Nature Communications.

Reviewer #1 (Remarks to the Author):

This study by the group of Shan addresses the question of how cells differentiate between proteins that must be exported from the cell versus those that must remain inside the cell. Signaling sequences, SRP, and SRP receptor are all necessary components for targeting these proteins to the ER, but they are not sufficient. With these components, even nascent proteins that lack signal sequences will tend to be targeted to the ER above the level observed in the cell. In this study, Shan and co-workers demonstrate that the NAC complex is also needed to further differentiate between these two classes of proteins. And that the most likely mechanism by which this occurs is through the co-binding of both NAC and SRP to ribosomes, with NAC tending to switch SRP into a non-functional state when nascent proteins are present that lack signal sequences. And that this reduces the affinity of the ribosome complex for the SRP receptor, thereby reducing improper protein targeting to the ER. This is an excellent study, an important new discovery, and advances our understanding of how the ensemble of protein biogenesis factors act together to lead to proper protein maturation.

The only major concern I have is with the assumption in the mathematical model that K_d for SRP is constant at all nascent chain lengths. Experiments from a number of groups show that K_d is length dependent – it starts high at short nascent chain length, goes low, then increases again at the longest lengths. How will incorporating this behavior affect the results? This should be addressed. (Note well, varying K_d as they did in panel 6f is not sufficient, as I understand that here they still kept the assumption of K_d independent of nascent chain length.)

We agree that the SRP-RNC K_d could vary based on the nascent chain (NC) length. According to the measurements by Flanagan et al. 2003, the SRP-RNC K_d for pPL is 8.4 nM at a NC length of 35 aa (before signal sequence emergence from the ribosome), 1.68 nM at a NC length of 66 amino acids, and 0.08–0.21 nM when the nascent chain is 84–220 amino acids long. We have re-run the modeling of the targeting reaction on RNC(ss) with the SRP-RNC K_d for short NCs raised by 15- and 100-fold relative to the K_d values for longer NCs (new Supplementary Fig. 9). The modeling yielded largely the same results as those assuming a length-independent SRP-RNC K_d . This is because the SRP-RNC K_d

is >100 fold below *in vivo* SRP concentration (400-500 nM). Thus, a fair amount of variations in K_d can be tolerated as they do not substantially affect the amount of RNC bound by SRP. Only when the SRP-RNC K_d rises above 200 nM will the targeting reaction begin to be affected, but such high K_d 's have never been observed.

This has been explained in both the Results (p16: “We found that the predicted targeting efficiencies remained largely the same when the concentration of SRP or the kinetics and affinity of RNC-SRP binding ($k_{on,SRP}$ and $K_{d,SRP}$) were varied by 1-2 orders of magnitude within their estimated limits (Fig. 7e-g). The modeling result is also insensitive to <100-fold changes in RNC-SRP K_d in response to nascent chain length (Supplementary Fig. 9). This strongly suggest that the *in vivo* SRP concentration is saturating with respect to its RNC binding affinity and that SRP-RNC binding is not rate-limiting for the overall targeting reaction.”) and Discussion (p20-21: “the pathway can tolerate perturbations of up to two orders of magnitude in SRP concentration, RNC-SRP binding affinity and/or kinetics. This can be attributed to the relatively high concentration of SRP *in vivo* compared to the K_d of RNC-SRP binding, and emphasizes that the specificity of the SRP pathway cannot be maintained solely by differences in the binding affinity of SRP to different RNCs.”)

I have the following minor suggestions:

1. The use of the ‘simulation’ is not appropriate when describing there solutions to the mathematical model. Rather, they ‘numerically solved’ the model. When you have an analytical function that you solve without any stochastic technique then it is not a simulation, the only uncertainty comes the numerical method itself. For example, with the authors technique, for the same arguments you always get the same solution. But for Monte Carlo, or Molecular dynamics you get different results. Examples of how the text can be altered, line 396: “to be simulated” -> “to be determined”; 399: “The simulation showed” -> “The model exhibits”, etc.

We corrected the description by replacing the terms “simulation” with “modeling” and “simulated” with “calculated”, “determined” or “modeled” where applicable.

2. A perspective on the interplay of timescales during protein biogenesis was written last year that should be cited PMID: 29414517 as it provides a broader context that the current manuscript fits into.

We have cited the paper as suggested.

Reviewer #2 (Remarks to the Author):

Review of Hsieh et al. “A ribosome-associated chaperone enables substrate triage in a co-translational protein targeting complex”

The manuscript of Hsieh et al. is a deep, rigorous investigation of a key question in protein biogenesis—how are proteins co-translationally targeted to the ER by

the SRP-receptor system with appropriate specificity? The authors tackle a key unanswered question in eukaryotic protein targeting—what determines the specificity for polypeptides for a signal-sequence versus those that do not; the SRP-receptor-signal peptide interactions on their own do not explain the observed *in vivo* targeting specificity. A possible co-factor in establishing fidelity of targeting is the abundant ribosome-binding chaperone NAC, which was thought to compete with SRP to increase specificity for appropriate signal sequence targeting. Here the authors upend this notion, and show conclusively that NAC binds (slightly anticooperatively) to ribosome-nascent chain-SRP complexes, and modulate the association rates of the complexes for the SRP receptor (RS), especially disfavoring the binding of ribosomes with short or non-signal peptides. The authors carry out careful kinetic targeting experiments using a combination of biochemical approaches (targeting kinetics in translation extracts with liver microsomes, and in particular careful bulk and single-molecule fluorescence assays to monitor SRP-ribosomal nascent chain (RNC) interaction, NAC-SRP-RNC interaction, and RS-SRP interactions, using a variety of nascent chain sequences and lengths. The results are rate and equilibrium constants for the key interactions and thermodynamic coupling parameters for NAC and SRP (they are negatively coupled, meaning binding of one makes binding of the partner less stable). They then perform challenging single-molecule FRET experiments to monitor the conformation of the SRP on an RNC, using doubly-labeled SRP. These experiments showed that SRP bound to a correct signal sequence RNC has largely a single conformation with high FRET, and addition of NAC shifts this conformational distribution, whereas with an incorrect signal sequence, the FRET distributions are broad (3 gaussian fit) and NAC shifts these distributions almost entirely into a single low FRET state. The authors interpret these results to show how NAC prepares an SRP-RNC complex for proper targeting, preventing the targeting to RS for incorrect or too short signal sequences. They support this model through a kinetic scheme and simulation, using their experimental data. The model is consistent with *in vivo* investigations, and should stimulate a number of additional *in vivo* and biophysical experiments. This is challenging biophysics of the first order, and as such the manuscript should be published in Nature Communications, but first I would like to see the authors address several points.

1. It would be nice to confirm co-occupancy of NAC and SRP. Have they performed direct FRET experiments to show that both factors reside on the ribosome? Their indirect data are supportive, but direct demonstration would be better.

Due to the lack of structural information on the ternary complex, we did not pursue the FRET experiment as suggested. Instead, we opted for a single-molecule TIRFM based colocalization experiment to directly observe NAC and SRP co-bound to the same RNC immobilized on the microscope slide. As predicted, we observed colocalization of NAC and SRP on both RNC(ss) and RNC(ssmt). In some co-binding events, we also observed FRET between NAC

and SRP on RNC(ss) as indicated by the anti-correlation between the Donor and Acceptor emission channels. These results are reported in new Fig. 5 and a new section in the main text (*"Detection of the RNC-NAC-SRP ternary complex"*).

2. For the single-molecule FRET data, are the timescales of the conformational changes for the mutant signal sequence, or NAC-induced changes? Are these rapid, or can state transitions be observed in the data. The authors should at least discuss this. Also, what is the molecular interpretation of these different states? Are their hints from structural data? I'm not sure that panel 5e is needed; it was confusing on first glance

For the labeling positions we chose, only the RNC(ss) +NAC and RNC(ssmt) - NAC samples had significant populations in different FRET states that can potentially have FRET transitions. Unfortunately, there were frequent photo-blinking on the ~10s timescale that interrupts the fluorescence traces, preventing us from determining the true dwell time. Besides the photo-blinking intervals, we did not see significant number of transitions between FRET states, indicating that these conformational states are kinetically stable.

The high FRET state in our single-molecule data is consistent with the RNC-SRP structure (Vorhees et al. 2015) in which SRP54 NG-domain docks at uL23 and is proximal to SRP19. We have now explained this in the Results (*"The FRET efficiency in this state is consistent with the cryoEM of the RNC•SRP, in which the SRP54-NG domain docks at ribosomal proteins uL23/uL29 near the exit tunnel and is in close proximity to SRP19. The population of SRP in the high FRET state strongly correlates with the activation of SRP-SR association kinetics, indicating that this conformation of SRP is optimal for SR binding¹⁰).*

The structural interpretation of medium and low FRET population is less clear. These states most likely arise from dynamic movement of the SRP54 NG-domain, which is tethered to the SRP54 M-domain by a flexible linker. As the SRP-SR binding kinetics correlates with the population of the high FRET state, the medium and low-FRET states are less competent for SR binding.

Figure 5e (new Fig. 6e) plots the cumulative population distribution of the different FRET states. This is a standard way to report the statistical analysis of the smFRET data. We have revised the legend of Figure 5e to make it clearer.

3. What is the dissociation rate of NAC from the ribosome? Was it determined independently, or we just have equilibrium constants?

We reported some of the NAC dissociation rate constants from the ternary complex in the new colocalization experiment in the new Figure 5. A complete characterization of NAC binding/dissociation kinetics under different conditions and for different RNCs is part of our ongoing work and beyond the scope of this paper.

4. The final model figure (7) should be improved. I had a difficult time to see the

differences in the nascent peptides from real ss to weak, to too short. Some suggestions--Perhaps thicken the lines for the peptides, and do not use the gray shaded background. Also, it would be nice to have the reaction arrows for all three states show explicitly or the molecules grouped. It looks like only the central strong sequence is undergoing the reaction in the no NAC case.

We have incorporated the reviewer's suggestions in our model figure.

Reviewer #3 (Remarks to the Author):

In their report Hsieh et al have studied the influence of the nascent-chain associated complex (NAC) on the fidelity of the co-translational protein targeting pathway by the Signal recognition particle (SRP). The authors conclude that the mammalian SRP and its receptor SR are unable to provide the required specificity for protein targeting to the ER membrane. Rather, they propose that by binding to SRP, NAC regulates the SRP-SR interaction and thus prevents the targeting of ribosome-nascent chain complexes (RNCs) with incorrect or short signal sequences. Thus, NAC would serve a crucial role in maintaining targeting fidelity.

The conclusion that NAC is a modulator of SRP specificity is actually not really new and has been already proposed earlier based on biochemical assays (Zhang et al., 2012) and on ribosome profiling data (del Alamo et al., 2011). However, considering the fact that the cellular NAC concentration equals the ribosome concentration, while SRP is largely sub-stoichiometric (approx. 1/100), it appears rather unlikely that the function of NAC is restricted to modulating the SRP function on the ribosome.

We extensively discussed previous literature that showed that NAC is a modulator of ER targeting specificity in the Introduction. However, the molecular mechanism underlying this regulation was unclear, and previous studies have mostly focused on potential effects of NAC on SRP or Sec61p binding to ribosomes. As discussed by the other two reviewers, we conclusively show here that NAC and SRP can co-bind on RNC and moreover, changes in SRP binding does not significantly affect targeting efficiency. Instead, the key mechanism by which NAC modulates SRP specificity is to alter SRP conformation and thus regulate SRP-SR association. Both aspects have been overlooked until this work.

Nowhere in the manuscript did we propose that modulating SRP specificity is the only role of NAC. We stated that NAC have other important functions given its high in vivo concentration (p4, "NAC is present at similar abundance as ribosomes in the cytosol^{16,21}, associates with a variety of translating ribosomes²², and can crosslink to the NC when the latter is still inside the ribosome exit tunnel²³. Deletion of NAC causes synthetic protein aggregation phenotype with the deletion of another cotranslational chaperone, Ssb, in yeast²⁴ and is lethal in higher eukaryotes^{25,26}, implicating it in cotranslational nascent protein folding"); these other roles should be addressed in future studies. However, this neither exclude the model we propose nor

invalidate the importance of NAC in modulating SRP specificity.

The initially suggested competitive binding model to which the authors refer (Wiedmann et al., 1994), which had suggested that NAC would prevent binding of incorrect cargos to the Sec61 complex at the ER membrane, was also already questioned by several studies (e.g. Raden & Gilmore, 1998; Neuhof et al., 1998). What is indeed correct and what has recently been further validated is that SRP and NAC engage almost identical binding sites on the ribosome and that they are both able to protrude into the ribosomal tunnel (Jomaa et al. 2016; Gamerdinger et al., 2019).

What the reviewer said “SRP and NAC engage almost identical binding sites on the ribosome and that they are both able to protrude into the ribosomal tunnel” is precisely the observations that led to the hypothesis that NAC and SRP compete with each other for binding to the ribosome and thus improve the specificity of SRP. In this work, we have conclusively showed that this is not the case using SRP-RNC, SRP-NAC binding assays and NAC-SRP single-molecule colocalization experiments. Instead, NAC co-binds with SRP and regulates SRP specificity allosterically.

We remind the reviewer that SRP has multiple binding sites on the ribosome. This and the presence of flexible linkers in SRP54 means that multiple alternative conformations of SRP-bound on the ribosome could be expected, as visualized in the smFRET data. The same is likely true for NAC, for which multiple crosslinking sites on the ribosome have been reported. It is therefore not structurally unreasonable that NAC and SRP can co-bind on the same ribosome, with each making a subset of their ribosome contacts.

Nevertheless, the authors data further support the hypothesis that NAC modulates the SRP-dependent targeting and they conclude that NAC ‘remodels the conformational landscape of SRP on the ribosome’. Although this is possibly true, the statement itself is rather vague and does not reveal any molecular mechanism. Thus, in summary, the manuscript contains interesting and rather complex data, but it does not really answer the question of how NAC improves targeting fidelity by the SRP pathway.

By “remodels the conformational landscape of SRP on the ribosome”, we specifically refer to single-molecule FRET experiments that show that, on ribosomes without a functional signal sequence, NAC eliminates the proximal conformation of SRP optimal for SR binding. This provides a molecular mechanism to explain the change in SRP-SR association kinetics. As we explained in response to comment 2 from Reviewer 2, the proximal conformation of SRP detected by smFRET is consistent with the structure of RNC-bound SRP in which SRP54-NG docks at uL23.

The reviewer seems to have a different concept for ‘mechanism’ than we or the other reviewers do. We provided multiple layers of mechanisms for the regulatory effect of NAC in this manuscript, ranging from rigorously defining the molecular step at which NAC exerts regulation (SRP-SR assembly kinetics), excluding the competitive model of SRP/NAC, to directly demonstrating conformational

regulation of ribosome-bound SRP by NAC. Moreover, we provide a valuable set of equilibrium and kinetic constants for multiple steps in the targeting reaction that enabled a complete mathematical modeling of the SRP pathway. All of these are significant mechanistic advances.

In addition, I have several issues the authors need to address for improving their study.

1. The authors use pre-prolactin (pPL) with different engineered signal sequences as substrates for their assays. Fig. 1c shows in vitro translocation of these substrates into microsomes and different to what the authors state, pPL containing the synthetic signal sequence (ss) is less efficiently translocated than pPL containing the wild type signal sequence. Yet, pPL containing the synthetic signal sequence is used in all experiments as reference. The rationale for this is not clear to me and the use of an already 'suboptimal' substrate as positive control is likely causing some bias in the interpretation. Therefore, the use of wild type pPL as positive control would be a much better choice.

The engineered signal sequence showed robust translocation well above the negative controls (ssmt and ssmt2) and serves as a proper positive control. It is also unclear what is the "bias" the reviewer is referring to. We also note that pPL has a highly optimized signal sequence with multiple consecutive leucines, which is unusual. The hydrophobic core of most co-translationally targeted substrates are much less ideal (Ast et al, Cell 2013).

2. lane 137/138: the authors state that SRP rejects non-optimal substrates. This statement is not correct. All the assay shows is that signal sequence cleavage of the ss, ssmt and ssmt2 constructs is reduced. Whether this is due to rejection by SRP, by the Sec61 complex or by signal peptidase cannot be deduced from these assays and would require more detailed analyses.

We wrote *"Thus, the SRP pathway effectively rejects nascent proteins with a weak or no signal sequence in a complete cell lysate."* By "SRP pathway", we mean the entire co-translational protein targeting pathway including the step at which the SRP-SR complex transfers the RNC to Sec61p.

3. The assays make use of a soluble SRP receptor and as the authors state, the absence of the transmembrane domain of SR β has only little effect on cell growth or protein translocation in yeast cells (Ogg et al., 1998). However, whether this also applies to the mammalian system is not known. The available data rather indicate that data from yeast cannot simply be transferred to the mammalian system without further proof. This is exemplified by the fact that neither SRP nor NAC are essential in yeast (or cells can at least cope with their absence), while both proteins are strictly essential in mammalian cells. It is also important to

emphasize that later work had shown that even in yeast the absence of the transmembrane domain impairs gating of the Sec61 complex (Jiang et al., 2007) and that contact between SR β and Sec61 is important for SRP release from the signal sequence (Fulga et al., 2001). Thus, the use of a soluble SR in an assay that is supposed to monitor membrane targeting is likely causing some additional bias.

We have shown here (Figure 1) and previously (Lee et al. 2018) that the soluble mammalian SR mediates efficient protein targeting to ER microsomes. The soluble SR is sufficient for binding to SRP, and the structure of the RNC-SRP-SR has been solved using the same soluble SR (Kobayashi et al. 2018). For this paper, which is focused on how NAC modulates SRP-RNC binding and SRP-SR association rate, the soluble SR is sufficient. We recognize that SR TMD may have important functions in interacting with Sec61, but this is beyond the scope of this study.

4. Lane154: as mentioned above, rejection is not shown by the assays depicted in Fig.1 and the ssmt2 construct has a much stronger effect on SR-SRP complex formation (Fig. 2). This demonstrates that the SRP system has an intrinsic proof-reading activity even in the absence of any additional factor, like NAC. This is also obvious from the data shown in 2g.

The rejection of ssmt and ssmt2 by the co-translational targeting pathway is clearly shown by the absence of the translocated product in Figure 1.

Ssmt2 is essentially the sequence of an intrinsically disordered cytosolic protein fragment devoid of any hydrophobic residues; it shows the maximal intrinsic discriminative capacity of SRP-SR without additional factors. However, native nascent chain sequences are not always as ideal as ssmt2 for SRP to reject. SRP also needs to distinguish the ER targeting signal from membrane proteins destined to other organelles, such as mitochondria, or stretches of hydrophobic residues that form the hydrophobic core of folded cytosolic proteins. These nascent chains, like ssmt1, have more minor differences with bona fide SRP substrates than ssmt2. In addition, the discrimination against both ssmt1 and ssmt2 during SRP-SR association improved >10-fold upon the addition of NAC (Fig. 2g), which further supports that NAC can improve the specificity of SRP.

5. Lane 197-199: This statement is misleading because the experimental system the authors are using cannot differentiate between early and late steps of the targeting reaction. SRP-SR interaction in vivo is a later step of targeting but the authors use the SRP-SR interaction as read-out for substrate recognition by SRP, which occurs before. Therefore, if the authors are indeed interested in understanding the fidelity of SRP to recognize its substrates, then monitoring the SRP-SR interaction is not the best read-out and monitoring SRP-RNC interaction directly as shown in Fig. 3 is a much better way.

We specifically monitored the SRP-RNC binding step in Figure 3 and the SRP-SR association step in Figure 2. These data conclusively showed that NAC does not help SRP discriminate between RNCs at the RNC-SRP binding step, and that the major step where NAC enhances SRP fidelity is during SRP-SR association.

6. Fig. 3 shows that SRP binds with nanomolar affinity to both RNC(ss) and RNC(ssmt). Here it would be important to see also the affinity for RNC(ssmt2), because the latter is indeed an incorrect substrate and should show similar nanomolar affinity if SRP indeed requires NAC for correct substrate recognition/rejection of incorrect substrates.

As we have stated in response to point 4, we do not think ssmt2 is strongly dependent on NAC for rejection by SRP. Ssmt1 represents a subset of incorrect substrate with more minor differences to bona fide SRP substrates and is dependent on NAC for rejection. We also note that Flanagan et al has measured the binding of SRP to RNC(globin), another cytosolic protein, and reported a K_d value of ~8 nM (Flanagan et al. 2003)

7. Why does NAC reduce the SRP-RNC(ss) interaction, if the ss signal sequence is considered to reflect a correct substrate (Fig. 3). This supports my earlier concern about using ss and these experiments should be performed with constructs harboring the native signal sequence.

As we showed in Figure 7 and explained in response to reviewer 1's comments, SRP-RNC binding is not the major discriminative step in the pathway, and the SRP pathway can tolerate substantial variations in the RNC-SRP K_d . The fraction of ribosome bound by SRP is, roughly, $[SRP]/(K_d + [SRP])$; this value is not sensitive to changes in K_d if $[SRP] \gg K_d$, which is the case here.

As this reviewer pointed out earlier, NAC and SRP share overlapping binding sites on the ribosome, one right at the exit site and one at uL23. It is reasonable that when both are bound to RNC, they need to adjust their position on the ribosome, which leads to an increase in K_d . As shown in previous structural work and in the smFRET analysis, ss stabilizes a conformation of SRP in which the SRP54-NG domain docks at uL23, which could overlap with NAC binding at/near the same site, whereas with RNC(ssmt), SRP is more flexible and adopts multiple other conformations on the ribosome, and therefore has less antagonism with NAC.

8. Lane 258: I am not sure how to understand the statement 'SRP and NAC modestly weaken the binding of each other to the RNC'. 'Modest weakening' does not seem to be compatible with the main conclusion of the manuscript that NAC enables substrate triage.

I also think that a competitive binding model cannot be excluded. Considering that both SRP and NAC protrude into the ribosomal tunnel in non-translating ribosomes or at very early stages of translation (Jomaa et al., 2016, Gernerding et al., 2019, Denks et al., 2017; Mercier et al., 2019), competition for ribosome binding can still occur at early stages but could then be followed by cooperative interaction when significant portions of the nascent chain have emerged from the ribosomal tunnel.

We have explained in response to comments 5 and 7 above that NAC increases discrimination against ssmt during the SRP-SR binding, rather than the RNC-SRP binding step.

We emphasize that the competitive binding model is conclusively excluded by the data here. The binding data of both NAC and SRP (Figures 3-4) cannot be explained by a competitive model. Single molecule colocalization experiments directly detected NAC and SRP bound to the same ribosome, and never detected SRP dissociation upon NAC binding (Figure 5). NAC cannot possibly change the conformation of SRP if it cannot co-bind with SRP on the same ribosome (Fig. 6).

We did all the binding measurements using RNCs with a defined nascent chain length. Therefore, the experimental results could not be explained by a combination of competition at one length and cooperative binding at another length. The new Supplementary Fig. 8 further showed that NAC does not weaken SRP binding to the empty ribosome, for which NAC could insert into the exit tunnel as described by the reviewer.

9. Fig. 4 shows that NAC cannot differentiate between RNC(ss) and RNC(ssmt). Again, it would be important to show the ssmt2 RNCS for analyzing whether NAC has any ability to discriminate between SRP and non-SRP signal sequences.

This is the same point as those in point 4 and point 6, and our response is the same.

10. Fig. 4a is also not properly designed: while it shows the decrease in donor fluorescence it does not show the corresponding increase in acceptor fluorescence, therefore the data cannot be properly interpreted.

The original Fig. 4a was obtained with 100-fold excess of NAC over SRP, so only 1% of NAC would FRET with SRP, and acceptor spectral changes would not be detectable. We have replaced Figure 4a with spectra that contain lower concentrations of NAC, so that the increase in acceptor fluorescence can be more readily seen.

11. The smFRET approach is nice, but what is the biological significance behind the low, medium and fast FRET populations? Also why is the medium-FRET

population less capable of interacting with SR? As these data stand right now, they don't provide much insight into the process, but are rather descriptive on a very fancy level.

As described in response to comment 2 from Reviewer 2, the high FRET state corresponds to the structure in which the SRP54-NG domain is docked at uL23. Previous work (Lee et al. 2018) showed that SRP by itself (without RNC or other additional factors) is in a medium FRET conformation and has very slow SRP-SR association rate ($<100 \text{ M}^{-1}\text{s}^{-1}$). Binding of the 80S ribosome and then signal sequence lead to increasing sampling of the high FRET state, which correlates with increasing rates of SRP-SR assembly. Thus, the high FRET conformation of SRP is optimized for SR binding. Building on this prior observation, our smFRET data now directly show that NAC inhibits SRP from adopting the high FRET conformation on RNC(ssmt), thus providing an explanation for the reduced SRP-SR interaction kinetics. This model was described in the Discussion (p19, “NAC significantly reduced the proximal conformation of SRP bound to RNC(ssmt) (Fig. 6). In addition, NAC substantially reduced the FRET efficiency between the nascent chain and SRP54-NG in the RNC(ssmt)-SRP complex (Fig. 3d). Both observations suggest displacement of SRP54-NG from its original docking site and are consistent with the reduced crosslink between SRP54 and signalless nascent chains on the upon addition of NAC^{27,28}. As both SRP54-NG and NAC dock near ribosomal protein uL23 at the exit tunnel, these results are most simply explained by a model in which NAC binding displaces SRP-NG from uL23 while the remainder of SRP remains bound to the ribosome.”).

Why this conformation of SRP is most conducive to SR binding is beyond the scope of this work and need to be addressed by separate structural studies.

12. Lane 354: different to what the authors state, signalless RNCs were not tested only RNCS with incorrect signal sequences.

We have defined “signalless ribosome” as “RNC without a functional signal sequence for ER targeting” (p4).

13. For addressing whether NAC influences the pre-emptive targeting of RNCs harboring short nascent chains, it would be much better to monitor the direct SRP-RNC interaction as in Fig. 3 (putting the label on the ribosome) instead of testing the rather indirect SRP-SR interaction. This would directly determine whether SRP binding to very short RNCs or even non-translating ribosomes is influenced by NAC

We emphasize that the observed suppression of pre-emptive targeting is due to the reduction in SRP-SR association rate at shorter nascent chain length. As shown in the response to Reviewer 1, even increasing the K_d between SRP and RNC by ~100-fold for shorter chain RNCs does not significantly decrease pre-

emptive targeting (Supplementary Fig. 9), and NAC would not be able to affect preemptive targeting by changing these binding constants. Moreover, the new Supplementary Fig. 8 shows that SRP binds to empty ribosomes with a K_d of ~20 nM, and NAC did not weaken SRP-ribosome binding affinity.

REVIEWER COMMENTS

Reviewer #1 (Remarks to the Author):

The authors have not satisfactorily addressed my primary concern. Specifically, they use the K_d versus nascent chain length trend observed by Flanagan and co-workers (2003), in which K_d is monotonically decreasing as a function of nascent chain length. This informed their input parameters for their model (Supplementary Fig. 9). Peter Walter observed with single molecule experiments that K_d is a non-monotonic function with nascent chain length (PMID: 24808175). The authors need to test the robustness of the conclusions they draw to this behavior in K_d with length, and show the results in the SI.

If the authors think Walter's data is erroneous such that they can reject the trends observed by that lab, they need to clearly articulate in the main text why they trust Flanagan's trends over Walter. Even in this case they should still report in the reviewer response what they find using this K_d trend.

Until this is done I cannot support publication.

Reviewer #2 (Remarks to the Author):

In the revised manuscript, the authors have addressed my concerns about NAC/SRP co-occupancy by performing additional single molecule experiments, and have now more deeply interpreted the single-molecule FRET data for SRP states. As such, the manuscript is now acceptable for publication in Nature Comm.

Reviewer #3 (Remarks to the Author):

In their revised manuscript Hsieh et al have addressed some of my previous concerns (mainly in the rebuttal letter, rather than in the manuscript/data itself), but there are still issues that in my opinion should be addressed by either including additional data or by explicitly stating experimental limitations of their experimental system in the manuscript. I also still think that the authors need to tone down some of their sometimes rather bold statements.

1. I still don't understand why the authors use pPL with a mutated signals sequence. In contrast to their response, pPL(ss) is not a robust SRP substrate. This is what their data actually show (Fig. 1D): translocation of pPL(ss) is reduced by about 50% compared to wild type. If the authors believe that wild type pPL contains a rather unusual signal sequence (statement in the rebuttal), then why did they use it in their previous studies (e.g. Lee et al., PNAS 2018)?

My concern about a possible 'bias' in their interpretation (comment 1 in my first review) refers to the following: The authors conclude that NAC is required to ensure/modulate the specificity of the SRP targeting system. The authors show this for pPL with a non-native signal sequence (which reduces translocation of pPL by 50%). But is this also observed for pPL with the wild type signal sequence? Without this control, all the authors can conclude that NAC is involved in modulating SRP recognition of mutated/sub-optimal signal sequences. This would also be an important conclusion, but then NAC would only be required for a subset of SRP substrates not for all SRP substrates in general.

2. lane 135/136: The statement that 'the SRP pathway effectively rejects nascent proteins' needs to be toned down because it is misleading and is not supported by the data it refers to. As it reads, it leaves the impression that SRP rejects nascent proteins with suboptimal signal sequences. In their rebuttal letter, the authors state that by SRP pathway, they 'mean the entire co-translational protein targeting pathway including the step at which the SRP-SRP complex transfers the RNC to Sec61p.' If this is their definition of SRP pathway, then they should share this definition with the reader. They

should also clarify that all they can conclude from the data at display is that signal sequence cleavage is reduced.

3. I don't question that the soluble SR can work in in vitro translocation assays or in their FRET experiments. This the authors have shown in their previous data and they show it here also. Nevertheless, I am not aware of any data clearly demonstrating that the soluble SR works as efficient and reliable as the native membrane-integral SR. If these data exist, they should be referenced and discussed. Without these data, the authors cannot exclude that the effect of NAC is caused by using a non-natural soluble SR. This is in particular important because the authors conclude that NAC enhances SRP fidelity at the SRP-SR association step.

4. The authors use the term 'signalless ribosome' for RNC without functional signal sequence. The authors need to be more precise and need to clearly differentiate between experiments in which they use non-functional signal sequences and those where they use substrates with no signal sequence at all.

Dear editor,

Thank you for your time and work handling this manuscript. We also thank the reviewers for their constructive comments, which helped making this manuscript stronger. Below are our point-by-point responses (blue) to the reviewers' comments (in black). We hope that the revised manuscript is now suitable for publication in Nature Communications.

Reviewer #1 (Remarks to the Author):

The authors have not satisfactorily addressed my primary concern. Specifically, they use the K_d versus nascent chain length trend observed by Flanagan and co-workers (2003), in which K_d is monotonically decreasing as a function of nascent chain length. This informed their input parameters for their model (Supplementary Fig. 9). Peter Walter observed with single molecule experiments that K_d is a non-monotonic function with nascent chain length (PMID: 24808175). The authors need to test the robustness of the conclusions they draw to this behavior in K_d with length, and show the results in the SI.

If the authors think Walter's data is erroneous such that they can reject the trends observed by that lab, they need to clearly articulate in the main text why they trust Flanagan's trends over Walter. Even in this case they should still report in the reviewer response what they find using this K_d trend.

Until this is done I cannot support publication.

We modeled protein targeting based on the K_d trend from Noriega *et al.* (2014), as suggested by the reviewer. We used the K_d values in Figure 5D of this paper to construct two parameterized K_d curves with different sensitivities to nascent chain length (new Supplementary Fig. 9d), neither of which significantly changes the outcome of the modeling (new supplementary Figure 9e and 9f). The results are discussed in the main text (p16).

Reviewer #2 (Remarks to the Author):

In the revised manuscript, the authors have addressed my concerns about NAC/SRP co-occupancy by performing additional single molecule experiments, and have now more deeply interpreted the single-molecule FRET data for SRP states. As such, the manuscript is now acceptable for publication in Nature Comm.

We thank the reviewer for constructive comments.

Reviewer #3 (Remarks to the Author):

In their revised manuscript Hsieh et al have addressed some of my previous concerns (mainly in the rebuttal letter, rather than in the manuscript/data itself), but there are still issues that in my opinion should be addressed by either including additional data or by explicitly stating experimental limitations of their experimental system in the manuscript. I also still think that the authors need to tone down some of their sometimes rather bold statements.

1. I still don't understand why the authors use pPL with a mutated signals sequence. In contrast to their response, pPI(ss) is not a robust SRP substrate. This is what their data actually show (Fig. 1D): translocation of pPL(ss) is reduced by about 50% compared to wild type. If the authors believe that wild type pPL contains a rather unusual signal sequence (statement in the rebuttal), then why did they use it in their previous studies (e.g. Lee et al., PNAS 2018)? My concern about a possible 'bias' in their interpretation (comment 1 in my first review) refers to the following: The authors conclude that NAC is required to ensure/modulate the specificity of the SRP targeting system. The authors show this for pPL with a non-native signal sequence (which reduces translocation of pPL by 50%). But is this also observed for pPL with the wild type signal sequence? Without this control, all the authors can conclude that NAC is involved in modulating SRP recognition of mutated/sub-optimal signal sequences. This would also be an important conclusion, but then NAc would only be required for a subset of SRP substrates not for all SRP substrates in general.

To ensure that pPL(ss) did not bias our conclusions, we measured SRP-SR association using RNC containing the wildtype pPL nascent chain (wt). As shown in the new Figure 2f, the k_{on} values with RNC(wt) are similar to those with RNC(ss) with or without NAC present. The role of NAC in maintaining the specificity during SRP-SR assembly remains the same if we were to compare pPL(wt) and pPL(ssmt). We described the new results in the main text.

2. lane 135/136: The statement that 'the SRP pathway effectively rejects nascent proteins' needs to be toned down because it is misleading and is not supported by the data it refers to. As it reads, it leaves the impression that SRP rejects nascent proteins with suboptimal signal sequences. In their rebuttal letter, the authors state that by SRP pathway, they 'mean the entire co-translational protein targeting pathway including the step at which the SRP-SR complex transfers the RNC to Sec61p.' If this is their definition of SRP pathway, then they should share this definition with the reader. They should also clarify that all they

can conclude from the data at display is that signal sequence cleavage is reduced.

We have rephrased this part in the main text: “The absence of signal sequence cleavage product with pPL(ssmt) and pPL(ssmt2) indicates that SRP-dependent cotranslational translocation effectively rejects nascent proteins with a weak or no signal sequence in a complete cell lysate.” We hope this would clearly convey what is experimentally observed and the conclusion we draw on.

3. I don't question that the soluble SR can work in *in vitro* translocation assays or in their FRET experiments. This the authors have shown in their previous data and they show it here also. Nevertheless, I am not aware of any data clearly demonstrating that the soluble SR works as efficient and reliable as the native membrane-integral SR. If these data exist, they should be referenced and discussed. Without these data, the authors cannot exclude that the effect of NAC is caused by using a non-natural soluble SR. This is in particular important because the authors conclude that NAC enhances SRP fidelity at the SRP-SR association step.

The balance of published data support soluble SR $\alpha\beta\Delta$ TM as a reasonable mimic of wildtype SR *in vivo*, in *in vitro* translocation, and in the SRP-SR GTPase cycle. Ogg et al, 1998 (ref 35) showed that the TMD of SR β is dispensable for its function in yeast, whereas its GTPase domain is essential. Fulga et al, 2001 (ref 36) showed that soluble mammalian SR $\alpha\beta\Delta$ TM is “functional in translocation”, and also concluded that GTP binding to SR β rather than its TMD is required for protein translocation. Finally, Mandon et al, 2003 (ref. 12) showed that soluble SR is comparable to wildtype SR-proteoliposome in the activated GTPase reaction with SRP in the presence of the ribosome. Note that the stimulated GTPase activity requires productive interactions between SRP and SR.

Although the Jiang et al, JCB 2007 paper suggested a role of the SR TMD in gating the Sec61 complex, this step occurs *after* SRP-SR binding and is beyond the scope of this work. Structurally, the SR β TMD is separated from the NG-domain of SR (which binds SRP) by the GTPase domain of SR β , the X-domain of SR α , and the ~200 amino acid long disordered linker in SR α . It is very unlikely that this TMD affects the SRP-SR binding step studied here.

We added the following to the main text (p6) where we introduced the solubilized SR mutant: “These measurements used a soluble SR $\alpha\beta\Delta$ TM lacking the nonessential N-terminal TMD of SR β (abbreviated as SR). The soluble SR can support yeast growth³⁵, is fully functional in supporting protein translocation (Fig. 1)³⁶, and displays the same activated GTPase reaction with SRP as wildtype SR¹².”

4. The authors use the term `signalless ribosome' for RNC without functional signal sequence. The authors need to be more precise and need to clearly differentiate between experiments in which they use non-functional signal sequences and those where they use substrates with no signal sequence at all.

The signal sequence that was used in each experiment is clearly defined (Figure 1b) and labeled in every figure panel. By the reviewer's definition, ssmt is a non-functional signal sequence and ssmt2 is without any signal sequence. Figure 1 and Figure 2 contain results with both ssmt and ssmt2, and all other experiments were done comparing ss to ssmt.

REVIEWERS' COMMENTS

Reviewer #1 (Remarks to the Author):

The authors have addressed my concern. I recommend publication.

Reviewer #3 (Remarks to the Author):

In their revised manuscript, the authors have addressed my previous concerns and I support publication in Nature communications.